# Robust Deepfake Detection Using Disjoint Ensembles on Redundant Features

## Abstract

Detecting GAN-generated deepfake images remains an open problem. Current detection methods fail against an adversary who adds imperceptible adversarial perturbations to the deepfake to evade detection. We propose **D**isjoint **D**eepfake **D**etection (D3), a detector designed to improve adversarial robustness beyond de facto solutions such as adversarial training. D3 uses an ensemble of models over disjoint subsets of the frequency spectrum to significantly improve robustness. Our key insight is to leverage a redundancy in the frequency domain and apply a saliency partitioning technique to disjointly distribute frequency components across multiple models. We formally prove that these disjoint ensembles lead to a reduction in the dimensionality of the input subspace where adversarial deepfakes lie. We then empirically validate the D3 method against white-box attacks and black-box attacks and find that D3 significantly outperforms existing state-of-the-art defenses applied to GAN-generated deepfake detection.

## 1 Introduction

Significant advances in deep learning are responsible for the advent of "deepfakes", which can be misused by bad actors for malicious purposes. Deepfakes broadly refer to digital media that has been synthetically generated or modified by deep neural networks (DNNs). Modern DNNs such as generative adversarial networks (GANs) (Goodfellow et al., 2014) are now capable of synthesizing hyper-realistic deepfakes, which can then be used to craft fake social media profiles (Martineau, 2019), generate pornography (Cole, 2017), spread political propaganda, and manipulate elections.

While recent work has made respectable efforts towards solving the deepfake detection problem, many of these detectors are rendered ineffective by *adversarial examples*. The deepfake detection problem asks the defender to classify a given image as deepfake or real (we focus on GAN-generated deepfake images). State-of-the-art detectors often leverage DNNs on the frequency space of the input image (Frank et al., 2020). However, Carlini & Farid (2020) (amongst others) have shown that such defenses fail — the adversary can simply use adversarial perturbation techniques to evade detection (Szegedy et al., 2014; Goodfellow et al., 2015; Biggio et al., 2013; Carlini & Wagner, 2017). Defending against adversarial examples, in general, has been shown to be a difficult task (Athalye et al., 2018), and is a critical problem in the deepfake detection setting.

Our key intuition to mitigate this problem is to utilize *redundant information in the frequency feature space* of deepfakes to generate *disjoint ensembles* for deepfake detection. Specifically, we show in Section 3.1 that we can achieve good detection performance with only a subset of the features, particularly in the frequency domain. This enables us to build an ensemble of performant classifiers, each using a disjoint set of frequencies. In contrast to traditional ensembles (where each model shares the same set of frequencies), a key advantage of this design is that non-robust frequencies are partitioned across all the models in the ensemble. Thus, an adversary is no longer able to perturb a single non-robust frequency to evade all models — rather, they must find perturbations to evade multiple sets of disjoint frequencies, which raises the attack cost.

Leveraging the above intuition, our key contributions are as follows:

1. We propose D3, a GAN-generated deepfake detection framework designed to be adversarially robust. D3 builds a robust ensemble of models that use disjoint partitions of the input

features. This is achieved by leveraging redundancy in the feature space. D3 achieves robustness while still exhibiting natural deepfake detection AUC-ROC scores as high as 99% (see Section 4.4 for details).

2. Extending the theoretical results by Tramèr et al. (2017b) on dimensionality of adversarial subspaces, we prove new bounds on the maximum number of adversarial directions that can be found under an ensemble with disjoint inputs. Our bounds are tight for both the $\ell_2$ and $\ell_\infty$ perturbation norms (Lemmas 3.1 and 3.2 in Section 3.3) and indicate that D3 reduces the dimension of the adversarial subspace.

3. We evaluate D3 against white-box and black-box attacks on a variety of GAN-generated deepfake images and find that D3 significantly outperforms state-of-the-art defenses such as ADP (Pang et al., 2019), GAL (Kariyappa & Qureshi, 2019) and DVERGE (Yang et al., 2020), suggesting a reduction in dimension of the adversarial subspace. For example, as indicated by our evaluation in Section 4.2, D3 maintains 100% adversarial accuracy against AutoAttack (Croce & Hein, 2020b) where baselines drop below 20%. Increasing the number of attack steps to 1000 reduces D3's robustness to 61% in comparison to 0% for all baselines.

## 2 BACKGROUND AND RELATED WORK

**Notation.** We consider a distribution $\mathcal{D}$ over $\mathcal{X} \times \mathcal{Y}$, where $\mathcal{X} \subseteq \mathbb{R}^d$ is the input space and $\mathcal{Y} \subseteq \mathbb{Z}^c$ is the finite class-label space. We denote vectors in boldface (e.g., $\mathbf{x}$). We denote a trained classifier as a function $\mathcal{F} : \mathcal{X} \to \mathcal{Y}$ (the classifier is usually parameterized by its weights $w$, omitted for brevity). We denote the loss function as $\mathcal{L}(\mathbf{x}, y)$. An ensemble classifier is a function $M_{(\mathcal{F}_1, \mathcal{F}_2, ..., \mathcal{F}_n)} : \mathcal{X} \to \mathcal{Y}$ that combines the logit outputs $l_1, l_2, ..., l_n$ of multiple classifiers $\mathcal{F}_1, \mathcal{F}_2, ..., \mathcal{F}_n$ with a voting aggregation function $\mathcal{A} : \mathbb{R}^{n \times c} \to \mathcal{Y}$.

We now provide a definition for adversarial examples against a classifier. For a classifier $\mathcal{F}$ and input-label pair $(\mathbf{x}, y)$, an adversarial example is a perturbed input $\mathbf{x}'$ such that (1) $\mathbf{x}'$ is misclassified, i.e., $\mathcal{F}(\mathbf{x}') \neq y$ and (2) $||\mathbf{x} - \mathbf{x}'||$ is within a small $\epsilon$ ball, where $||.||$ is a given norm. The value of $\epsilon$ is chosen to be small so that the perturbation is imperceptible to humans.

**Deepfake Detection.** The research community has made rapid progress towards detecting GAN-generated deepfake images. Examples of proposed detection schemes include DNN classifiers (Wang et al., 2020; Yu et al., 2019), color-space anomaly detection (McCloskey & Albright, 2018), and co-occurrence matrix analysis (Nataraj et al., 2019), amongst others (Tariq et al., 2019; Mi et al., 2020; Marra et al., 2018; Guarnera et al., 2020; Marra et al., 2019).

However, a recent line of work has significantly advanced the state-of-the-art for deepfake detection by leveraging *frequency-space analysis*. For example, Frank et al. (2020) proposed the idea of detecting deepfakes with the Discrete Cosine Transform (DCT) as a pre-processing transform before a binary DNN-based deepfake classifier. Similar work has also achieved remarkable performance — Zhang et al. (2019) use GAN simulators to extract similar frequency artifacts, and Durall et al. (2019) successfully train DNNs in the frequency domain to detect deepfakes.

Unfortunately, the aforementioned detectors have been rendered ineffective in adversarial settings. Specifically, Carlini & Farid (2020) showed that frequency-based detectors are vulnerable to adversarial examples — an adversary can add imperceptible adversarial perturbations to a deepfake that evade such detectors, rendering them ineffective. Others have corroborated this observation (Hussain et al., 2021; 2022b; Neekhara et al., 2021; Gandhi & Jain, 2020; Vo et al., 2022; Shahriyar & Wright, 2022; Fernandes & Jha, 2020; Liao et al., 2021; Hussain et al., 2022a).

**Detecting Adversarial Deepfakes.** A widely accepted solution to countering adversarial examples is to train the model on adversarial examples, generated during training (Madry et al., 2018). However, Carlini & Farid (2020) suggest that adversarial training alone is unlikely to achieve significant improvement in robustness in the difficult deepfake detection setting. Our experiments in Section 4.2 confirm that Frank et al. (2020)'s frequency-space deepfake detector, even when adversarially trained, cannot withstand deepfake examples crafted using a variety of attacks (see Tables 1 and 2). While we also adversarially train each model in our ensembles, we find that our ensembles significantly improve robustness over a standalone adversarially trained model (see Section 4.2).

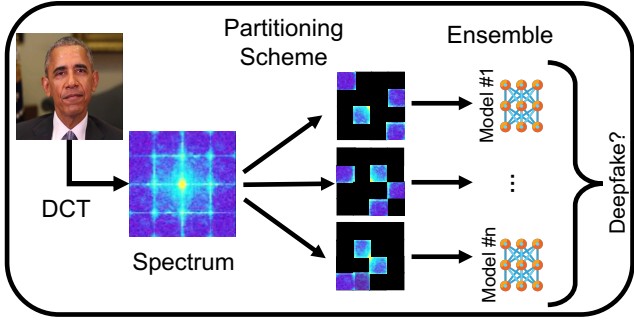

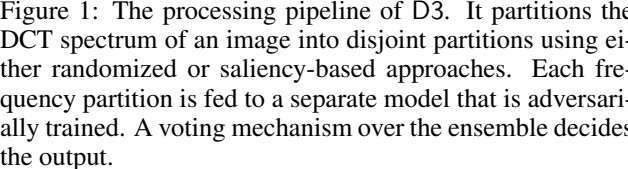

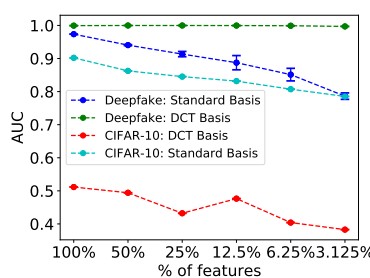

Figure 1: The processing pipeline of D3. It partitions the DCT spectrum of an image into disjoint partitions using either randomized or saliency-based approaches. Each frequency partition is fed to a separate model that is adversarially trained. A voting mechanism over the ensemble decides the output.

Figure 2: AUC-ROC scores of a single CNN classifier trained on a fixed subset (randomly selected) of the input features. Redundancy in frequency spectra permits near-perfect deepfake detection even when using $\sim 3\%$ of the components (see green line).

Another class of defenses focuses on removing the effect of the adversarial perturbation without any changes made to the underlying detection technique — unfortunately, these approaches are computationally intensive, e.g., upto 30 minutes per image (Gandhi & Jain, 2020; Jiang et al., 2021), in comparison to no additional overhead in D3. Nevertheless, both approaches are complementary in that pre-processing could be combined with adversarially robust deepfake detectors.

Another approach is an ensemble of models (Pang et al., 2019; Kariyappa & Qureshi, 2019; Yang et al., 2021) — in principle, the adversary is then forced to attack multiple models, instead of just one. However, He et al. (2017) have shown that arbitrarily ensembling models does not necessarily lead to more robust defenses. One reason for this (as per Ilyas et al. (2019)) is that each model tends to learn the same non-robust features, i.e., an adversary is able to perturb the same feature to evade all models. Recent efforts at adversarially robust ensembles have attempted to mitigate this phenomenon by introducing additional regularizers to the training process (Pang et al., 2019; Kariyappa & Qureshi, 2019; Yang et al., 2021). In contrast, D3 bypasses this phenomenon by disjointly partitioning features amongst models, thereby outperforming these approaches (see Section 4.2).

More recently, Dutta et al. (2021) proposed EnsembleDet, which ensembles different DNN architectures on the same set of features as a defense. We differ from EnsembleDet philosophically, in that we do not require different architectures, and instead leverage artifact redundancy to design frequency-partitioned ensembles (see Section 3 for a more detailed explanation). Furthermore, D3's partitioning approach to achieve disjoint ensembles is novel, and it offers theoretical and empirical advantages where such feature redundancy is available, e.g., GAN-generated deepfakes.

## 3 OUR APPROACH

We now present D3 (Figure 1), an ensemble deepfake detection framework that leverages disjoint frequency models to achieve robust detection of GAN-generated deepfake images. Section 3.1 presents our observation of redundant information in the frequency space of deepfakes. Section 3.2 details how redundancy allows frequencies to be partitioned between multiple models for robust ensemblinga nd explains our exact frequency partitioning schemes. Finally, Section 3.3 provides theoretical insights into why D3 improves adversarial robustness.

### 3.1 REDUNDANT INFORMATION IN DEEPFAKE IMAGE SPECTRA

As discussed in Section 2, ensembles are a promising approach to adversarial robustness, so long as perturbing the same set of features does not simultaneously evade the individual models. We propose designing an ensemble that avoids this shortcoming by *disjointly partitioning the input feature space* amongst individual models.

Unfortunately, for traditional image classifications tasks like CIFAR10 (Krizhevsky et al., 2009) and ImageNet (Deng et al., 2009), a disjoint partitioning is likely to hurt natural classification performance, as partitioning results in information loss for each model. We visualize this phenomenon for CIFAR10 via the red and teal lines in Figure 2, which plots the worsening performance of a simple convolutional classifier [1] using increasingly small subsets of the input features (pixel or frequency).

Our first key insight is that, unlike traditional image classification tasks, disjoint partitioning is feasible for deepfake detection. Specifically, we observe a "redundancy" in frequency-space artifacts — signals relevant for deepfake detection [2] are distributed throughout the frequency spectrum. This observation is best exemplified by the green line in Figure 2, which plots the deepfake detection performance using increasingly small subsets of the spectrum using the same classifier. Using as few as $\sim 3\%$ of the frequency components yields near-perfect deepfake detection performance. We emphasize that this does not hold for subsets of pixels (blue line), as signals for detection are not well-distributed in the RGB space. Thus, the frequency-space contains plenty of redundancy, which can potentially be leveraged to design a robust ensemble.

## 3.2 LEVERAGING REDUNDANCY TO BUILD A ROBUST ENSEMBLE

Using the observations in Section 3.1, one can craft a robust ensemble by partitioning the frequency components amongst multiple detector models, without hurting natural detection performance. Figure 1 visualizes this partitioning as part of our ensembling pipeline. Specifically, for each individual model we mask (i.e., zero out) the frequencies not used. For example, consider an ensemble of two such "disjoint" models $F_A$ and $F_B$, where the full-spectrum frequency feature vector $f = [f_1; f_2]$. Then, the input feature vector to $F_A$ is $[f_1; 0]$ and to $F_B$ is $[0; f_2]$. Since the input feature space (frequencies) is not shared amongst the individual models, the adversary cannot simply attack the ensemble by targeting common frequencies.

Furthermore, we note that the choice of partitioning scheme, i.e., *how* the frequencies are partitioned plays an important role in robustness of the ensemble. Specifically, the chosen scheme should aim to design all models as "equals" — if some models are less robust than others, then the adversary can target them to overturn the ensemble's decision. Below we propose a naive solution (random partitioning), followed by a scheme designed to directly achieve this goal (saliency partitioning).

**Random Partitioning.** For an ensemble of size $n$, this naive partitioning scheme divides the set of $d$ frequency components into $n$ equally-sized disjoint subsets, uniformly at random. Each model in the ensemble then receives one subset.

**Saliency Partitioning.** While signals for deepfake detection are distributed throughout the spectrum, there still exists a *adversarial saliency* ordering of these frequencies that determines their robustness for the deepfake detection task. For an ensemble of size $n$, our saliency partitioning technique is aimed at ensuring each model receives a fair proportion of salient frequencies. To this end, we follow Carlini & Wagner (2017) and Feng et al. (2022) to compute saliency values for all frequencies. This is achieved by adversarially perturbing deepfake $x$ to $x + \delta^x$, where $\delta^x$ is the perturbation computed with an unbounded $l_\infty$ PGD attack until attack success or 1000 steps, and computing saliency $s_i$ for the $i^{th}$ frequency as

$$s_i = \mathbb{E}_{x \in \mathcal{X}} \nabla f(x + \delta^x)_i \cdot \delta_i^x \tag{1}$$

where subscript $i$ denotes the $i^{th}$ component. Intuitively, higher gradients and larger perturbation magnitudes imply larger saliencies. Frequencies are then ordered by their saliencies, and distributed in a round-robin fashion amongst the models.

## 3.3 ADVERSARIAL SUBSPACE OF DISJOINT ENSEMBLES

Given the partitioning approach in Section 3.2, we now show that an ensemble of such disjoint frequency models increases robustness against adversarial examples by reducing the dimension of the adversarial subspace. For a single model $\mathcal{F}$ and input $\mathbf{x}$, Tramèr et al. (2017b) approximate

---

[1]We use the same architecture for our primary defense implementation; details are provided in Section 4.1

[2] Frank et al. (2020) show that these artifacts manifest in the form of a grid-like spectral pattern, and attribute their presence to the upsampling process in generative models. Existing work proposes detectors that leverage the *entire frequency spectrum* for deepfake detection.

the $k-$dimensional adversarial subspace as the span of orthogonal perturbations $\mathbf{r_1}, \cdots, \mathbf{r_k} \in \mathbb{R}^d$ such that $\mathbf{r_i}^\intercal \mathbf{g} \geq \gamma \; \forall \; 1 \leq i \leq k$ where $\mathbf{g} = \nabla_\mathbf{x} \mathcal{L}_\mathcal{F}(\mathbf{x}, y)$, $\mathcal{L}_\mathcal{F}$ is the loss function used to train $\mathcal{F}$, and $\gamma$ is the increase in loss sufficient to cause a mis-classification. For perturbations satisfying the $\ell_2$-norm, i.e. $||\mathbf{r_i}||_2 \leq \epsilon \; \forall \; 1 \leq i \leq k$, the adversarial dimension $k$ is bounded by $\frac{\epsilon^2 ||\mathbf{g}||_2^2}{\gamma^2}$ (tight). In what follows, we extend this result and provide bounds for dimensionality of the shared adversarial subspace between $n$ disjoint models. We provide tight bounds for both $\ell_2$ and $\ell_\infty$ norms in Lemma 3.1 and Lemma 3.2 respectively (with detailed proofs in Appendix A.1). We consider these bounds for two voting mechanisms: (1) majority, where the ensemble outputs deepfake if at least $\lceil n/2 \rceil$ classifiers predict deepfake, and (2) at-least-one, where the classifier outputs deepfake if at least one classifier predicts deepfake, otherwise it outputs real.

**Lemma 3.1.** *Given $n$ disjoint models, $\mathcal{F}_1, ..., \mathcal{F}_n$, having gradients $\mathbf{g_1}, \cdots, \mathbf{g_n} \in \mathbb{R}^d$ for input-label pair $(\mathbf{x}, y)$ (where $\mathbf{g_j} = \nabla_\mathbf{x} \mathcal{L}_{\mathcal{F}_j}(\mathbf{x}, y)$), the maximum number $k$ of orthogonal vectors $\mathbf{r_1}, \mathbf{r_2}, \cdots, \mathbf{r_k} \in \mathbb{R}^d$ satisfying $||\mathbf{r_i}||_2 \leq \epsilon$ and, $\mathbf{r_i}^\intercal \mathbf{g_j} \geq \gamma_j$ for all $1 \leq j \leq n$ (at-least-one voting) or for at least $\lceil \frac{n}{2} \rceil$ models (majority voting), for all $1 \leq i \leq k$ is given by:*

$$k = \min \left( d, \left\lfloor \frac{\epsilon^2}{\left( \sum\limits_{j=1}^n \gamma_j \right)^2} \sum_{j=1}^n ||\mathbf{g_j}||_2^2 \right\rfloor \right) \quad \text{(at-least-one voting)} \tag{2}$$

$$k \leq \min \left( d, \left\lfloor \max_{|K| = \lceil \frac{n}{2} \rceil} \frac{\epsilon^2}{\left( \sum\limits_{j \in K} \gamma_j \right)^2} \sum_{j \in K} ||\mathbf{g_j}||_2^2 \right\rfloor \right) \quad \text{(majority voting)} \tag{3}$$

$$k \geq \min \left( d, \left\lfloor \min_{|K| = \lceil \frac{n}{2} \rceil} \frac{\epsilon^2}{\left( \sum\limits_{j \in K} \gamma_j \right)^2} \sum_{j \in K} ||\mathbf{g_j}||_2^2 \right\rfloor \right) \quad \text{(majority voting)} \tag{4}$$

**Lemma 3.2.** *Given $n$ disjoint models, $\mathcal{F}_1, ..., \mathcal{F}_n$, having gradients $\mathbf{g_1}, \cdots, \mathbf{g_n} \in \mathbb{R}^d$ for input-label pair $(\mathbf{x}, y)$ (where $\mathbf{g_j} = \nabla_\mathbf{x} \mathcal{L}_{\mathcal{F}_j}(\mathbf{x}, y)$), the maximum number $k$ of orthogonal vectors $\mathbf{r_1}, \mathbf{r_2}, \cdots, \mathbf{r_k} \in \mathbb{R}^d$ satisfying $||\mathbf{r_i}||_\infty \leq \epsilon$ and $\mathbb{E}[\mathbf{g_j}^\intercal \mathbf{r_i}] \geq \gamma_j$ for all $1 \leq j \leq n$ (at-least-one voting) or for at least $\lceil \frac{n}{2} \rceil$ models (majority voting), for all $1 \leq i \leq k$*

$$k = \min \left( d, \left\lfloor \min \left( \frac{\epsilon^2 ||\mathbf{g_1}||_1^2}{n^2 \gamma_1^2}, ..., \frac{\epsilon^2 ||\mathbf{g_n}||_1^2}{n^2 \gamma_n^2} \right) \right\rfloor \right) \quad \text{(at-least-one voting)} \tag{5}$$

$$k = \min \left( d, \left\lfloor \text{median} \left( \frac{\epsilon^2 ||\mathbf{g_1}||_1^2}{n^2 \gamma_1^2}, ..., \frac{\epsilon^2 ||\mathbf{g_n}||_1^2}{n^2 \gamma_n^2} \right) \right\rfloor \right) \quad \text{(majority voting)} \tag{6}$$

**Implications.** If all $n$ disjoint models in the ensemble are "near-identical" (as expected per our saliency partitioning scheme), i.e., $||\mathbf{g_1}||_2^2 \approx \cdots \approx ||\mathbf{g_n}||_2^2$ and $\gamma_1 \approx \cdots \approx \gamma_n$, then Lemma 3.1 for at-least-one voting reduces to $k \approx \min \left( d, \left\lfloor \frac{\epsilon^2 ||\mathbf{g_1}||_2^2}{n \gamma_1^2} \right\rfloor \right)$. This implies that an ensemble of $n$ disjoint models offers potential reduction in dimensionality of the adversarial subspace by a factor of $n$ compared to any individual constituent disjoint model. Similar interpretation holds for Lemma 3.2, where reductions are now by a factor of $n^2$.

Furthermore, using Tramèr et al. (2017a)'s GAAS technique for dimensionality estimation, we can show that individual disjoint models exhibit lower dimensionality as compared to that of a single model trained on the full frequency spectrum[3]. Additional empirical analysis of individual models in Section 4.3 confirms this. This implies that ensembling such disjoint models leads to even further reduction in dimensionality of the adversarial subspace.

---

[3]Intuitively, this can be attributed to each disjoint model receiving fewer non-robust features, but continuing to perform well due to the redundancy observation. Additional details for GAAS estimation can be found in Appendix A.2.

## 4 EXPERIMENTAL EVALUATION

Our experiments broadly aim to answer the following questions:

**Q1.** What is the performance of D3 ensembles against white-box and black-box adversaries?

**Q2.** How does choice of partitioning scheme and ratio impact robustness?

**Q3.** What is the natural deepfake detection performance of D3 ensembles vs. a full-spectrum detector on unperturbed deepfakes?

In the following sections, we address these questions by detailing our setup and experiments.

### 4.1 EXPERIMENTAL SETUP

We adopt the following settings to evaluate D3 for detecting deepfakes and adversarial deepfakes.

**Datasets and Pre-processing.** We primarily perform our experiments using a dataset comprising real, and GAN-generated deepfake images of individuals. The real images are obtained from Flickr-Faces HQ (FFHQ) dataset of celebrity headshots (Karras et al., 2019). Deepfake images are generated using the state-of-the-art StyleGAN, trained with standard hyperparameters on the 128x128 FFHQ. Specifically, deepfake images are generated using the de facto approach of sampling points uniformly on a $512-$dimensional sphere, and feeding them through the generator DNN to obtain images of "people that do not exist". From each of the classes (deepfake and real), we use 50,000 images for training, 10,000 for validation, and 10,000 for testing sets. Finally, we transform all deepfake images to their frequency space representation using the 2D-DCT function (Ahmed et al., 1974). We also evaluate D3 using other datasets, including images from BigGAN (Brock et al., 2018) (conditional GAN trained on ImageNet (Deng et al., 2009)), StarGAN (Choi et al., 2020) (image-to-image conditional GAN trained on CelebA-HQ (Karras et al., 2018)), and other types of non-GAN generated deepfakes, i.e., FaceSwap and NeuralTextures deepfakes from the FaceForensics++ dataset (Rössler et al., 2019). Additional details for these datasets can be found in Appendix A.3.3

D3 **Architecture and Training.** All models in our D3 ensembles implement the same shallow four-layer CNN binary classifier employed by the baseline Frank et al. (2020) frequency-space defense. All models are adversarially trained with PGD-10 attacks (Madry et al., 2018) on deepfake images, by minimizing the binary cross-entropy loss with the Adam optimizer and an initial learning rate of 0.001. We use a batch size of 512 for a maximum of 20 epochs (with early stopping based on AUC-ROC over the validation set). All models are implemented using the PyTorch v1.6.0 framework for Python. Additional details on adversarial training, including hyperparameters, can be found in Appendix A.3.1.

D3 **Configurations.** We consider two configurations of our ensemble that partition frequency components into 2 or 4 disjoint subsets, i.e., ensembles of size $n = 2$ and $n = 4$, respectively. We select these values of $n$ based on analysis presented in Section 4.3. We refer to the following configurations in rest of the paper: (a) D3-R(2) and D3-R(4) for $n = 2$ or $n = 4$ that use the random partitioning strategy; and (b) D3-S(2) and D3-S(4) for $n = 2$ or $n = 4$ that use the saliency partitioning strategy. Ensemble decisions are obtained via a voting function that computes the mean logits of $n$ models.

**Baselines.** We consider an adversarially trained (AT) version of Frank et al. (2020)'s full spectrum frequency-space classifier, as well as several ensemble-based defenses against adversarial examples: GAL (Kariyappa & Qureshi, 2019), ADP (Pang et al., 2019), DVERGE (Yang et al., 2020) as our baselines. TRS (Yang et al., 2021) performed poorly for deepfake detection - we weren't able to get good natural performance (likely due to the regularizer not converging over the frequency space). For all baselines, we denote the ensemble size in parentheses, e.g., ADP(4). Additional baseline details are provided in Appendix A.3.1.

**Metrics.** Since deepfake detection is a binary classification task, we use AUC-ROC score to measure the natural detection performance of D3. AUC-ROC measures the area under the receiver-operator curve, and compares the true positive rate to the false positive rate for different classifier thresholds. We employ *adversarial accuracy* (fraction of correctly classified adversarial examples) as our performance metric for robustness. A higher adversarial accuracy implies more robustness.

Table 1: Adversarial accuracies for $\ell_\infty$ perturbation attacks. We present accuracies for all D3 configurations, for the AT (adversarial training) baseline, and for ensemble-based baselines ADP, GAL, and DVERGE (DV). Parentheses denote the number of models used for defenses and the number steps used for attacks.

| Attacks | $\epsilon$ | AT(1) | ADP(4) | GAL(4) | DV(4) | D3-R(2) | D3-S(2) | D3-R(4) | D3-S(4) |
|---------|-----------|-------|--------|--------|-------|---------|---------|---------|---------|
| APGD-CE(50) | 1/255 | 7.6 | 1.0 | 0.2 | 28.0 | 98.2 | **100.0** | 99.8 | **100.0** |
| | 4/255 | 0.2 | 0.0 | 0.0 | 21.5 | 97.8 | **100.0** | 99.2 | **100.0** |
| | 8/255 | 0.0 | 0.0 | 0.0 | 21.5 | 97.2 | **100.0** | 98.4 | **100.0** |
| | 16/255 | 0.0 | 0.0 | 0.0 | 0.0 | 6.0 | 96.2 | 83.8 | **100.0** |
| FAB | 4/255 | **100.0** | 99.4 | 96.6 | 71.5 | 99.6 | 99.0 | **100.0** | 99.5 |
| | 16/255 | **100.0** | 99.4 | 72.8 | 69.0 | 99.6 | 99.0 | **100.0** | 99.5 |
| | 32/255 | 61.6 | 93.2 | 0.2 | 65.0 | 32.6 | 98.0 | **99.5** | **99.5** |
| | 64/255 | 0.0 | 4.8 | 0.0 | 0.0 | 0.0 | 5.8 | 0.0 | **9.5** |
| Square | 4/255 | 99.8 | 96.4 | 96.0 | 28.0 | 99.4 | 98.2 | **99.6** | 97.0 |
| | 16/255 | 92.0 | 66.6 | 51.0 | 28.0 | 96.8 | 98.2 | **99.4** | 96.6 |
| | 32/255 | 25.8 | 13.6 | 6.6 | 21.0 | 50.8 | 94.8 | 88.0 | **96.2** |
| | 64/255 | 0.0 | 0.4 | 0.0 | 0.5 | 0.4 | 45.2 | 4.0 | **95.0** |

Table 2: Adversarial accuracies for $\ell_2$ perturbation attacks for same configurations as in Table 1

| Attacks | $\epsilon$ | AT(1) | ADP(4) | GAL(4) | DV(4) | D3-R(2) | D3-S(2) | D3-R(4) | D3-S(4) |
|---------|-----------|-------|--------|--------|-------|---------|---------|---------|---------|
| APGD-CE(50) | 0.5 | 44.0 | 25.0 | 5.4 | 45.5 | 98.4 | **100.0** | 99.8 | **100.0** |
| | 1 | 16.0 | 1.8 | 1.8 | 24.5 | 97.2 | **100.0** | 98.9 | **100.0** |
| | 5 | 0.0 | 0.0 | 0.0 | 21.5 | 96.0 | **100.0** | 98.2 | **100.0** |
| | 10 | 0.0 | 0.0 | 0.0 | 0.0 | 1.2 | 76.2 | 63.6 | **100.0** |
| FAB | 10 | 99.4 | 99.6 | 39.6 | 67.5 | 97.6 | 99.0 | **100.0** | **100.0** |
| | 20 | 20.4 | 81.0 | 0.0 | 64.5 | 0.8 | **99.0** | 54.5 | 98.5 |
| | 40 | 0.0 | 0.0 | 0.0 | 0.0 | 0.0 | 0.0 | 0.0 | 0.0 |
| | 80 | 0.0 | 0.0 | 0.0 | 0.0 | 0.0 | 0.0 | 0.0 | 0.0 |
| Square | 10 | 99.0 | 94.4 | 94.0 | 13.0 | **100.0** | 99.0 | **100.0** | 99.4 |
| | 20 | 83.8 | 79.8 | 65.6 | 10.0 | 98.2 | 99.0 | **99.8** | 99.6 |
| | 40 | 33.2 | 32.0 | 6.4 | 7.5 | 56.4 | 97.0 | 86.2 | **99.6** |
| | 80 | 1.0 | 0.0 | 0.0 | 0.0 | 0.0 | 22.2 | 2.4 | **95.8** |

## 4.2 ROBUSTNESS AGAINST ADVERSARIAL EXAMPLES

We evaluate robustness of D3 and baselines against attacks in both the white-box and black-box settings, and find that D3 is more robust than baselines. All attacks are crafted using the AutoAttack benchmark, which is widely accepted as an evaluation standard for adversarial examples (Croce & Hein, 2020b). White-box attacks include the adaptive step-size PGD attacks using cross-entropy (APGD-CE) and Carlini-Wagner (APGD-CW) loss functions as well as the Fast Adaptive Boundary (FAB) attack. Black-box attacks include Square attack (Andriushchenko et al., 2020).

Table 1 presents adversarial accuracies under different $\ell_\infty$ perturbation budgets and find that D3 (specifically, the D3-S(4) configuration) significantly outperforms all baselines under white-box settings. For example, D3-S(4) achieves 99.5% for attack budgets of $\epsilon = 8/255$ vs. 21.5% for closest performing baseline DVERGE. This performance gap tends to widen under large perturbation budgets. Similar trends hold under the Square black-box attack; for example, D3 achieves 95.0% adversarial accuracy vs. 0.5% for the best baseline on a perturbation budget of $\epsilon = 64/255$.

Table 2 presents adversarial accuracies under different $\ell_2$ perturbation budgets, and we find a similar trend of D3 outperforming baselines by a significant margin (e.g., 100% accuracy vs. 21.5% for DVERGE at $\epsilon = 5$).

Table 3 present results for $\ell_2$ and $\ell_\infty$ attacks where we increase the number of iteration steps in the white-box APGD-CE attack. While D3's accuracy does drop to 61% under 1000 attack steps, it continues to outperform all baselines (which unanimously drop to 0%). Figure 3 demonstrates that above trends continue to hold under increased perturbation budgets.

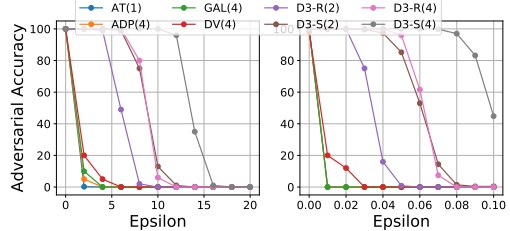 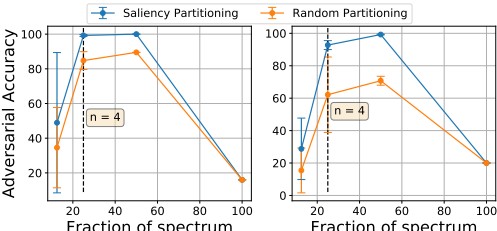

Figure 3: Adversarial accuracy of D3 and baselines as perturbation budget is increased ($\ell_2$ on the left, $\ell_\infty$ on the right).

Figure 4: Expected adversarial accuracy ($\ell_2$ on the left, $\ell_\infty$ on the right) of individual models at different partitioning ratios $\in [0, 100]$.

Table 3: Adversarial acc. for APGD-CE attack at $\ell_2 = 1$, $\ell_\infty = 8/255$ under increasing steps.

| | | | | $\ell_2$ | | | | |
|---|---|---|---|---|---|---|---|---|
| Steps | AT(1) | ADP(4) | GAL(4) | DV(4) | D3-R(2) | D3-S(2) | D3-R(4) | D3-S(4) |
| 50 | 16.6 | 2.5 | 1.5 | 24.5 | **100.0** | **100.0** | **100.0** | **100.0** |
| 500 | 0.0 | 0.0 | 0.0 | 21.5 | 63.0 | 98.6 | 99.0 | **100.0** |
| 1000 | 0.0 | 0.0 | 0.0 | 0.0 | 34.0 | 85.5 | 94.5 | **100.0** |
| | | | | $\ell_\infty$ | | | | |
| 50 | 0.0 | 0.0 | 0.0 | 21.5 | 98.0 | **100.0** | **100.0** | **100.0** |
| 500 | 0.0 | 0.0 | 0.0 | 18.5 | 0.0 | 2.0 | 49.0 | **92.0** |
| 1000 | 0.0 | 0.0 | 0.0 | 0.0 | 0.0 | 0.5 | 8.0 | **61.0** |

We also evaluate D3 on other GAN-generated datasets (BigGAN and StarGAN) and find a similar trend; it significantly outperforms baseline deepfake detection methods. Additional details can be found in Appendix A.3.3.

Finally, we find that naive attack strategies such as transfer attacks from a full-spectrum model do poorly against D3. Additional details can be found in Appendix A.3.2.

**Summary.** We find for Q1 that D3 ensembles offer significant robustness improvements against both white-box and black-box attacks crafted using the popular AutoAttack benchmark (Croce & Hein, 2020b) that tries multiple state-of-the-art attacks on models. For both $\ell_\infty$ and $\ell_2$ perturbation norms, D3 continues to exhibit high adversarial accuracies (e.g., $> 99.5\%$ for $\ell_\infty, \epsilon = 32/255$) where existing defenses fail. D3 continues to offer accuracies greater than 61% ($\ell_\infty, \epsilon = 8/255$) and even 100% ($\ell_2, \epsilon = 1$) when attack steps are increased to 1000, a setting where existing state-of-the-art defenses are reduced to 0% accuracy.

## 4.3    IMPACT OF PARTITIONING SCHEME ON ROBUSTNESS

In Section 4.2, we observed that ensembles constructed using our saliency partitioning strategy (D3-S(2) and D3-S(4)) demonstrated higher adversarial accuracies than those using our random partitioning strategy (D3-R(2) and D3-R(4)) under all attacks. Both strategies outperformed the full-spectrum detector. We explain these results by examining the robustness of the individual constituent models under both strategies.

Figure 4 plots the expected adversarial accuracy over the individual models in the ensemble for both schemes (using the same APGD-50 attack on 1000 images for $\ell_2 \epsilon = 1$, $\ell_\infty \epsilon = 4/255$). We observe that all the individual models (looking at 50% and 25%) are more robust than the full-spectrum (100%) detector, thus satisfying the conditions in Lemmas 3.1 and 3.2. For individual models looking at 12.5%, this is not the case (see high standard deviations). Additionally, individual models produced using saliency partitioning are more robust than those using random (due to proportionate distribution of robust/non-robust frequencies), suggesting further reduction in adversarial subspace

dimensionality when ensembling. Finally, arbitrarily partitioning beyond 25% does not improve robustness of individual models, suggesting diminishing gains beyond an ensemble of $n = 4$ models.

**Summary.** We find for Q2 that individual disjoint models obtained via saliency partitioning are significantly more robust than a full spectrum model. These improvements still exist for random partitioning, but are not as large. Further, individual models with partitioning ratios of 50% and 25% outperform a full spectrum trained model, with ratios beyond 25% not improving robustness.

### 4.4 NATURAL DEEPFAKE DETECTION PERFORMANCE

As per Table 4, we find that all D3 configurations (even D3-R(4) and D3-S(4) where individual models use 25% of frequency components) achieve AUC-ROC scores over 99%. In other words, there is no loss in natural performance when compared to the full spectrum frequency classifier Frank et al. (2020).

| Type | AT(1) | D3-R(2) | D3-S(2) | D3-R(4) | D3-S(4) |
|------|-------|---------|---------|---------|---------|
| Frequency | 99.99 | 99.96 | 99.98 | 99.93 | 99.95 |
| Pixel | 97.59 | 90.42 | 96.45 | 90.11 | 93.83 |

Table 4: Natural deepfake detection AUC-ROC scores of D3 vs full-spectrum detector AT(1).

We emphasize the importance of frequency-space redundancy in enabling our partitioning scheme, allowing D3 to achieve these scores. To this end, we also compute scores obtained from ensembles that do not apply DCT to images, i.e., D3 ensembles in the pixel space. Each of these ensembles exhibit considerably lower performances as artifacts are not evenly distributed throughout the pixel space. We refer the reader to Figure 2 to further visualize this effect.

**Summary.** We find for Q3 that D3 ensembles preserve the natural accuracy of a full-spectrum model — we observe reductions of less than 1% in AUC-ROC score. This comparable performance can be attributed to the redundancy of relevant artifacts in the frequency space. In contrast, ensembles with partitioning in the pixel space suffer from drops of 8% in scores.

## 5 LIMITATIONS AND FUTURE WORK

**Societal impact and limitations.** Modern deepfakes raise several societal and security threats; D3 is a step towards mitigating that. Nonetheless, adversarial deepfakes also have benign use cases, e.g., anonymization of an end-user on an online network; D3 could prevent such anonymization. Additionally, D3 is focused on GAN-generated deepfake images — since it relies upon redundancy in the frequency space, it may not be effective against against other types of deepfakes that avoid these artifacts. We believe that the benefits of D3 outweigh such potential concerns.

**Applicability to domains other than deepfake detection.** We presented D3 as a ensemble framework for adversarially robust deepfake detection. However, we hypothesize that this approach could be applied to any classification task that exhibits redundancy in a feature space. While we are unaware of such a space for the popular CIFAR10 and ImageNet classification tasks (see Table 9 in Appendix A.3.4 for preliminary results on CIFAR10 using the DCT), there are several classification tasks in, say, the audio domain that exhibit redundancy in features, e.g., keyword spotting and fake speech detection. Exploring this hypothesis is an interesting future research direction.

## 6 CONCLUSIONS

In this paper, we present D3, an ensemble approach to deepfake detection that exploits redundancy in frequency feature space by partitioning the frequencies across multiple models. We show theoretical advantages to such disjoint partitioning of input features, in that it reduces the dimensionality of the adversarial subspace. We empirically validate that D3 offers significant gains in robustness under multiple adaptive white-box and black-box attacks, maintaining high adversarial accuracy where best-performing baselines drop below 20%.

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

# A  APPENDIX

In Appendix A.1, we provide proofs for our bounds on dimensionality of the adversarial subspace of disjoint ensembles, such as D3. Appendix A.2 empirically estimates the dimension of the adversarial subspace using a combinatorial construction of orthogonal perturbations. In Appendix A.3, we provide details of our baseline configurations and attack settings for reproducibility purposes, as well as additional experimental results on other datasets. Finally, in Appendix A.5, we provide additional insight and motivation for our saliency frequency partitioning scheme.

## A.1  PROOFS

### A.1.1  ORTHOGONAL GRADIENTS

Prior to providing proofs for the adversarial dimensions, we demonstrate that gradients for disjoint classifiers are always orthogonal to each other. We will use this for our later results. Given input space $\mathcal{X}$ and class-label space $\mathcal{Y}$, we have $n$ disjoint classifiers $\mathcal{F}_1, ..., \mathcal{F}_n$. If $T$ is the DCT transformation matrix, we can define $T_i$ to be the transformation matrix for the classifier $\mathcal{F}_i$. Each disjoint transformation $T_i$ has a lot of zeros. Only the rows corresponding to the unmasked frequencies of classifier $\mathcal{F}_i$ have non-zero entries. Moreover, since no frequency is shared by any two classifiers, the $j^{th}$ row will have non-zero entries in exactly one of the $n$ disjoint transformation matrices, i.e. $T_i T_j^{\mathsf{T}} = O \ \forall i \neq j$.

Next, the $n$ disjoint classifiers $\mathcal{F}_1, ..., \mathcal{F}_n$, where $\mathcal{F}_i : T_i \mathbf{x} \to y$, are trained using loss functions $\mathcal{L}_{\mathcal{F}_1}, ..., \mathcal{L}_{\mathcal{F}_n}$ respectively. Now, the dot product between the gradients of classifiers $\mathcal{F}_i$ and $\mathcal{F}_j$ is given by

$$
\begin{aligned}
(\nabla_{\mathbf{x}} \mathcal{L}_{\mathcal{F}_i})^{\mathsf{T}} \left( \nabla_{\mathbf{x}} \mathcal{L}_{\mathcal{F}_j} \right) &= (T_i^{\mathsf{T}} \nabla_{T_i \mathbf{x}} \mathcal{L}_{\mathcal{F}_i})^{\mathsf{T}} \left( T_j^{\mathsf{T}} \nabla_{T_j \mathbf{x}} \mathcal{L}_{\mathcal{F}_j} \right) \\
&= (\nabla_{T_i \mathbf{x}} \mathcal{L}_{\mathcal{F}_i})^{\mathsf{T}} T_i T_j^{\mathsf{T}} \left( \nabla_{T_j \mathbf{x}} \mathcal{L}_{\mathcal{F}_j} \right) \\
&= 0
\end{aligned}
\tag{7}
$$

### A.1.2  PROOF OF LEMMA 3.1

From Tramèr et al. (2017b), we know that for a classifier $\mathcal{F} : \mathcal{X} \to \mathcal{Y}$ where $\mathcal{X} \in \mathbb{R}^d$ is the input space and $\mathcal{Y}^c \in \mathbb{Z}$ is the finite class label space, the dimension of the adversarial subspace around input-label pair $(\mathbf{x}, y)$ where $\mathbf{x} \in \mathcal{X}$ and $y \in \mathcal{Y}$, is approximated by the maximal number of orthogonal perturbations $\mathbf{r_1}, \mathbf{r_2}, ..., \mathbf{r_k}$ such that $||\mathbf{r_i}||_2 \leq \epsilon$ and $\mathbf{g}^{\mathsf{T}} \mathbf{r_i} \geq \gamma \ \forall \ 1 \leq i \leq k$. Here,

$\mathbf{g} = \nabla_{\mathbf{x}} L(\mathcal{F}(\mathbf{x}), y)$ and $\gamma$ is the increase in loss function $L$ sufficient to cause a mis-classification. Tramèr et al. (2017b) provide a tight bound for $k$:

$$k = \min\left(d, \left\lfloor \frac{\epsilon^2 ||\mathbf{g}||_2^2}{\gamma^2} \right\rfloor\right) \tag{8}$$

We now extend this result for $n$ disjoint classifiers. Let $\mathbf{g}' = \frac{\sum_{j=1}^{n} \mathbf{g_j}}{n}$.
Now, for *at-least-one voting*,

$$\mathbf{g}'^{\mathsf{T}} \mathbf{r_i} = \frac{\sum_{j=1}^{n} \mathbf{g_j}^{\mathsf{T}} \mathbf{r_i}}{n} \geq \frac{\sum_{j=1}^{n} \gamma_j}{n} \quad \forall\, 1 \leq i \leq k \tag{9}$$

Applying the result from Tramèr et al. (2017b) (Equation 8) on the above inequality (Equation 9), we get:

$$\begin{aligned}
k &= \min\left(d, \left\lfloor \frac{\epsilon^2 n^2 ||\mathbf{g}'||_2^2}{\left(\sum_{j=1}^{n} \gamma_j\right)^2} \right\rfloor\right) \\
&= \min\left(d, \left\lfloor \frac{\epsilon^2 \sum_{j=1}^{n} ||\mathbf{g_j}||_2^2}{\left(\sum_{j=1}^{n} \gamma_j\right)^2} \right\rfloor\right). \quad \text{(since } \mathbf{g_i}^{\mathsf{T}} \mathbf{g_j} = 0 \;\; \forall i \neq j, \text{ using Equation 7)}
\end{aligned} \tag{10}$$

Now, for *majority voting*, we again apply the results from Tramèr et al. (2017b) (Equation 8). However, the derivation now depends on the selection of $\lceil \frac{n}{2} \rceil$ models that the adversary chooses to target. To obtain the lower and upper bounds, we can select $\lceil \frac{n}{2} \rceil$ with the most and least adversarial dimensions respectively. Following a similar derivation as before, we get :

$$k \geq \min\left(d, \left\lfloor \min_{|K|=\lceil \frac{n}{2} \rceil} \frac{\epsilon^2 \sum_{j=1}^{n} ||\mathbf{g_j}||_2^2}{\left(\sum_{j=1}^{n} \gamma_j\right)^2} \right\rfloor\right) \tag{11}$$

$$k \leq \min\left(d, \left\lfloor \max_{|K|=\lceil \frac{n}{2} \rceil} \frac{\epsilon^2 \sum_{j=1}^{n} ||\mathbf{g_j}||_2^2}{\left(\sum_{j=1}^{n} \gamma_j\right)^2} \right\rfloor\right) \tag{12}$$

### A.1.3 PROOF OF LEMMA 3.2

Follow up work from Tramèr et al. (2017a) also provides a tight bound for the adversarial dimension in the $\ell_\infty$ case. They provide a tight bound for the number of $k$ orthogonal perturbations $\mathbf{r_1}, ..., \mathbf{r_k} \in \mathbb{R}^d$ such that $||\mathbf{r_i}||_\infty \leq \epsilon$, given by $sign(\mathbf{g})^{\mathsf{T}} \mathbf{r_i} = \frac{\epsilon d}{\sqrt{k}} \; \forall 1 \leq i \leq k$ where $sign(\mathbf{g})$ is the signed gradient.

We now extend this result for $n$ disjoint classifiers. For $\mathbf{g}' = \frac{\sum_{j=1}^{n} \mathbf{g_j}}{n}$, since $\mathbf{g_j}'s$ are non-zero only on non-overlapping dimensions, we can see that $sign(\mathbf{g}')^{\intercal}r = \sum_{j=1}^{n} sign(\mathbf{g_j})^{\intercal}r \; \forall \mathbf{r} \in \mathbb{R}^d$. Applying the above results here, we get

$$\sum_{j=1}^{n} sign(\mathbf{g_j})^{\intercal}\mathbf{r_i} = \frac{\epsilon d}{\sqrt{k}} \; \forall 1 \leq i \leq k \tag{13}$$

Now, similar to Tramèr et al. (2017a), we compute the perturbation magnitude along a random permutation of the signed gradient. For each $1 \leq j \leq n$ and $1 \leq i \leq k$, we get :

$$\begin{aligned}
\mathbb{E}[\mathbf{g_j}^{\intercal}\mathbf{r_i}] &= \mathbb{E}\left[\sum_{p=1}^{d} |g_j^{(p)}| \cdot sign(g_j^{(p)}) \cdot r_i^{(p)}\right] \\
&= \sum_{p=1}^{d} |g_j^{(p)}| \mathbb{E}\left[sign(g_j^{(p)}) \cdot r_i^{(p)}\right] \\
&= \frac{\epsilon \|\mathbf{g_j}\|_1}{n\sqrt{k}}
\end{aligned} \tag{14}$$

## A.2 Estimating Dimension of Adversarial Subspace

In this section, we use Tramèr et al. (2017a)'s combinatorial construction to find $k$ orthogonal adversarial directions for a full spectrum (100%) model as well as individual models that use a fraction of the frequency features. The frequency features are selected using the saliency partitioning scheme (Section 3.2).

We construct orthogonal perturbations by multiplying the row vectors from a Regular Hadamard matrix component-wise with $sign(\mathbf{g})$ where $\mathbf{g}$ is the model gradient. We consider 3 settings, based on fraction of frequency features - 100%, 50% and 25%. For each setting, we then select 1000 correctly classified inputs, and compute loss gradients for each input. Using a regular Hadamard Matrix of order $w$, the above construction gives $w$ orthogonal directions. Out of these $w$, the number of directions that are actually adversarial provide an estimate for $k$. We run the analysis for $w \in \{4, 16, 36, 64, 100\}$, and select the maximum value.

Figure 5 plots the proportion of images (out of 1000) with at least $k$ adversarial dimensions. We observe that both the 50% and 25% settings have fewer images with high dimension of the adversarial subspace. These results support the analysis done in Section 3.3. Further, the reduced adversarial dimension for the 50% and 25% settings is consistent with their improved robustness over the full spectrum (100%) model (Section 4.3).

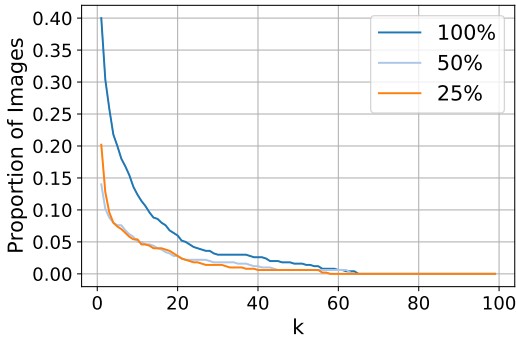

Figure 5: Distribution of adversarial saliency magnitudes for frequency components.

## A.3 ADDITIONAL EVALUATION DETAILS

### A.3.1 DETAILS FOR BASELINES

In this section, we provide additional details for all models/ensembles used for the baseline evaluation. We implement all the baselines using the four-layer CNN employed by Frank et al. (2020).

**Adversarial-Training (Madry et al., 2018).** We employ adversarial training, which involves training the CNN on adversarial examples for deepfake images only (not real images). Adversarial examples are constructed using the PGD-10 attack (i.e., 10 steps of gradient descent), with $\epsilon = 1/255$ and a step size of 0.001. We train for 20 epochs with a batch size of 512.

**GAL (Kariyappa & Qureshi, 2019).** This approach minimizes transferability via a regularizer. Specifically, GAL minimizes the cosine similarity between individual models' loss gradients. We set the coefficient for standard cross-entropy loss as 1, and for the cosine loss as 0.5. We train for 35 epochs with a batch size of 128.

**ADP (Pang et al., 2019).** This approach also minimizes transferability via a regularizer. The primary term in ADP's regularizer maximizes the volume spanned by individual model's non-prediction output vectors, i.e., the vectors obtained by removing the top-confidence class, so as to maximize diversity without affecting accuracy. We set coefficients for standard cross-entropy loss as 1, for the volume term in the regularizer as 0.5, and for the entropy term in the regularizer as 2. We train for 1 epoch with a batch size of 128.

**DVERGE (Yang et al., 2020).** DVERGE minimizes transferability by adversarially training individual models on "adversarial features" constructed against other models in the ensemble. Adversarial features are constructed by minimizing a novel feature-distillation objective over 10 steps, with $\epsilon = 0.07$ and step size as 0.007. We train for 5 epochs with a batch size of 128.

### A.3.2 DETAILS FOR ATTACKS

We consider the following attacks for our evaluation. These attacks constitute the entire ensemble of attacks used in the AutoAttack Benchmark. We tune the attacks where necessary to get the strongest attack setting.

**APGD-CE/CW (Croce & Hein, 2020b)** is a step-size free variant of the standard PGD attack. It adjusts the step size during the attack based on the convergence of the loss and the overall perturbation budget. We optimize the adaptive PGD-attack on the Cross Entropy (CE) and Carlini Wagner (CW) loss functions. We use the same set of parameters for the attack as mentioned in AutoAttack Benchmark other than the step size decay parameter $\alpha$ which we set to 0.1 instead of 2.

**FAB (Croce & Hein, 2020a)** is a iterative first order attack that utilizes geometry of the decision boundary to minimize the perturbation required to cause mis-classification. We use the same set of parameters as AutoAttack.

**Square (Andriushchenko et al., 2020)** is an efficient black-box attack that uses random square shaped updates to approximate the decision boundary. We use the same set of parameters as AutoAttack.

We also include in Table 5 results for a "baseline" attack, in which we evaluate D3 on adversarial examples computed against a full-spectrum classifier. As expected, such adversarial examples do not evade D3.

| | $\ell_2$ | | | | | $\ell_\infty$ | | | | |
|---|---|---|---|---|---|---|---|---|---|---|
| $\epsilon$ | 0.5 | 1 | 5 | 10 | 20 | 1/255 | 4/255 | 8/255 | 16/255 | 32/255 |
| Accuracy | 100.0 | 100.0 | 100.0 | 100.0 | 1.4 | 100.0 | 100.0 | 100.0 | 100.0 | 49.1 |

Table 5: Adversarial accuracy of D3-S(4) under adversarial examples crafted against a full-spectrum detector.

### A.3.3 EVALUATION ON OTHER DATASETS

**BigGAN.** The BigGAN dataset comprises images GAN-generated images belonging to the 1000 classes in the ImageNet (Deng et al., 2009) dataset. Images are generated using BigGAN (Brock et al., 2018) trained on the 256x256 ImageNet dataset of 1000 object and animal classes. Images are resized to 128x128. We use 35,000 images for training, 7,500 images for validation, and 7,500 for testing sets. Corresponding real images are obtained from the ImageNet dataset.

Table 6: BigGAN adversarial accuracies for $\ell_\infty$ and $\ell_2$ perturbation attacks for same configurations as in Table 2

| Attacks | $\epsilon$ | AT(1) | ADP(4) | GAL(4) | DV(4) | D3-S(4) |
|---------|-----------|-------|--------|--------|-------|---------|
| | | | $\ell_\infty$ | | | |
| APGD-CE(50) | 1/255 | 16.2 | 9.1 | 41.5 | 5.5 | **91.0** |
| | 4/255 | 14.4 | 9.1 | 9.4 | 0.0 | **90.0** |
| | 8/255 | 14.4 | 9.1 | 9.4 | 0.0 | **78.3** |
| | 16/255 | 14.4 | 9.1 | 9.4 | 0.0 | **26.6** |
| APGD-CW(50) | 1/255 | 15.9 | 9.1 | 2.3 | 5.4 | **91.0** |
| | 4/255 | 14.4 | 9.1 | 43.8 | 0.0 | **90.5** |
| | 8/255 | 14.4 | 9.1 | 9.4 | 0.0 | **79.6** |
| | 16/255 | 14.4 | 9.1 | 9.4 | 0.0 | **26.7** |
| | | | $\ell_2$ | | | |
| APGD-CE(50) | 0.1 | 23.6 | 9.2 | 59.4 | 13.6 | **91.0** |
| | 1 | 16.6 | 9.1 | 27.1 | 2.8 | **91.0** |
| | 5 | 14.4 | 9.1 | 9.4 | 0.0 | **70.5** |
| | 10 | 14.4 | 9.1 | 9.4 | 0.0 | **26.2** |
| APGD-CW(50) | 0.1 | 25.1 | 9.3 | 61.6 | 14.5 | **91.0** |
| | 1 | 15.4 | 9.1 | 27.4 | 3.1 | **91.0** |
| | 5 | 14.4 | 9.1 | 9.4 | 0.0 | **71.0** |
| | 10 | 14.4 | 9.1 | 9.4 | 0.0 | **26.2** |

**StarGAN.** The StarGAN dataset comprises images GAN-generated images belonging to the 1000 classes in the ImageNet (Deng et al., 2009) dataset. Images are generated using StarGAN (Choi et al., 2020) trained on the 256x256 CelebA-HQ dataset (resized from 1024x1024) of celebrity head-shots (Karras et al., 2018). Since StarGAN is an image-to-Image translation GAN that changes the "style", i.e., hairstyle, eye color, etc of an existing real image, we generate images using randomly sampled styles. Images are resized to 128x128. We use 21,000 images for training, 4,500 images for validation, and 4,500 for testing sets. Corresponding real images are obtained from CelebA-HQ.

**FaceForensics++ (DeepFakes and NeuralTextures.)** This dataset comprises the DeepFakes and NeuralTextures subsets of the FaceForensics++ dataset (Rössler et al., 2019). These are face-swap deepfake videos generated using neural, non-GAN techniques, and are thus not the primary focus of this work. Nonetheless, we include preliminary evaluation of natural performance on these videos. We extract faces from each video frame using the dlib library (King, 2009). Face images are resized to 128x128. We use 6,265 images for training, 1,342 for validation, and 1,343 for testing.

Other ensemble approaches such as EnsembleDet (Dutta et al., 2021) might work better on non-GAN deepfakes such as those from FaceForensics++, as they use different architectures, instead of relying upon redundancy. Unfortunately, source code for EnsembleDet is unavailable, and their official evaluation only considers attacks such as FGSM and BIM, which are known to be weaker than AutoAttack, making a direct comparison difficult.

### A.3.4 EVALUATION ON CIFAR10

Table 9 presents results for attacking the D3-S(4) ensemble and AT baseline, for the CIFAR10 classification task. While D3-S(4) offers some improvements at the smallest perturbation budgets, it quickly drops off. We suspect that a more carefully chosen feature space with redundancies for animal/vehicle classification would improve these results.

Table 7: StarGAN adversarial accuracies for $\ell_\infty$ and $\ell_2$ perturbation attacks for same configurations as in Table 2

| Attacks | $\epsilon$ | AT(1) | ADP(4) | GAL(4) | DV(4) | D3-S(4) |
|---|---|---|---|---|---|---|
| | | | $\ell_\infty$ | | | |
| APGD-CE(50) | 1/255 | 14.6 | 8.6 | 1.6 | 18.5 | **100.0** |
| | 4/255 | 0.0 | 0.0 | 0.0 | 3.0 | **99.6** |
| | 8/255 | 0.0 | 0.0 | 0.0 | 0.0 | **0.1** |
| | 16/255 | 0.0 | 0.0 | 0.0 | 0.0 | **0.1** |
| APGD-CW(50) | 1/255 | 11.5 | 17.4 | 2.3 | 19.3 | **100.0** |
| | 4/255 | 0.0 | 0.0 | 0.0 | 2.7 | **99.8** |
| | 8/255 | 0.0 | 0.0 | 0.0 | 0.0 | **0.1** |
| | 16/255 | 0.0 | 0.0 | 0.0 | 0.0 | **0.1** |
| | | | $\ell_2$ | | | |
| APGD-CE(50) | 0.1 | 45.3 | 87.5 | 13.5 | 25.3 | **100.0** |
| | 1 | 9.5 | 0.0 | 0.9 | 11.6 | **100.0** |
| | 5 | 0.0 | 0.0 | 0.0 | **2.7** | 0.1 |
| | 10 | 0.0 | 0.0 | 0.0 | **2.7** | 0.1 |
| APGD-CW(50) | 0.1 | 20.9 | 90.8 | 16.2 | 19.3 | **100.0** |
| | 1 | 3.3 | 0.1 | 0.3 | 5.8 | **100.0** |
| | 5 | 0.0 | 0.0 | 0.0 | **2.7** | 0.1 |
| | 10 | 0.0 | 0.0 | 0.0 | **2.7** | 0.1 |

| Dataset | AT(1) | D3-S(4) |
|---|---|---|
| DeepFakes | 98.8 | 90.6 |
| NeuralTextures | 94.6 | 80.8 |

Table 8: Natural accuracies of D3 and a full-spectrum classifier on non-GAN generated deepfakes from the FaceForensics++ dataset.

Table 9: CIFAR10 adversarial accuracies for $\ell_\infty$ and $\ell_2$ perturbation attacks for same configurations as in Table 2.

| Attacks | $\epsilon$ | AT(1) | D3-S(4) |
|---|---|---|---|
| | | $\ell_\infty$ | |
| APGD-CE(50) | 1/255 | 1.7 | **59.8** |
| | 4/255 | 0.1 | **1.2** |
| | 8/255 | **0.1** | 0 |
| | 16/255 | **0.1** | 0 |
| | | $\ell_2$ | |
| APGD-CE(50) | 0.5 | 15.1 | **63.5** |
| | 1 | 13.6 | **39.7** |
| | 5 | 0.9 | **8.2** |
| | 10 | 0.8 | **6.2** |

## A.4 PERTURBATION REQUIRED FOR SUCCESSFUL ATTACKS

## A.5 UNDERSTANDING FREQUENCY PARTITIONING

Recall that each model in our ensembles uses a subset of frequencies. However, as per Ilyas et al. (2019)'s robust features model, these frequencies can likely be categorized as either robust, or non-robust. We would like each model to receive an equal number of robust frequencies. In Section 3.2, we discussed two schemes for disjoint partitioning of frequency components — a more "naive" random partitioning, and our proposed saliency partitioning. To better visualize how saliency par-

| Unperturbed Deepfake | AT ($\ell_2$) | D3-S(4) ($\ell_2$) | AT ($\ell_\infty$) | D3-S(4) ($\ell_\infty$) |
|---|---|---|---|---|

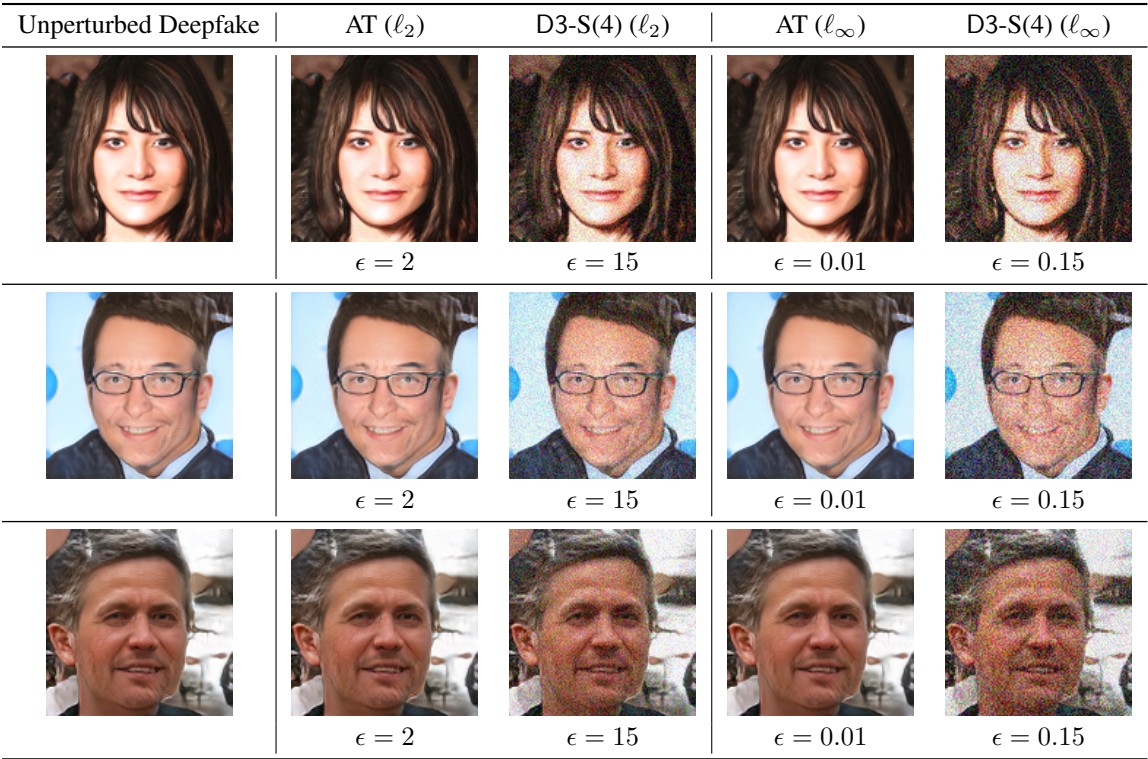

Figure 6: Minimum Perturbation required for successful attacks

titioning achieves our goal of equal distribution, Figure 7 plots a histogram of adversarial saliency magnitudes. Recall that these saliencies are computed using the gradient (and perturbation) magnitude *at the adversarial example*, i.e., our adversarial saliencies are a heuristic measure of "robustness" of each frequency component, with higher saliencies implying less robustness.

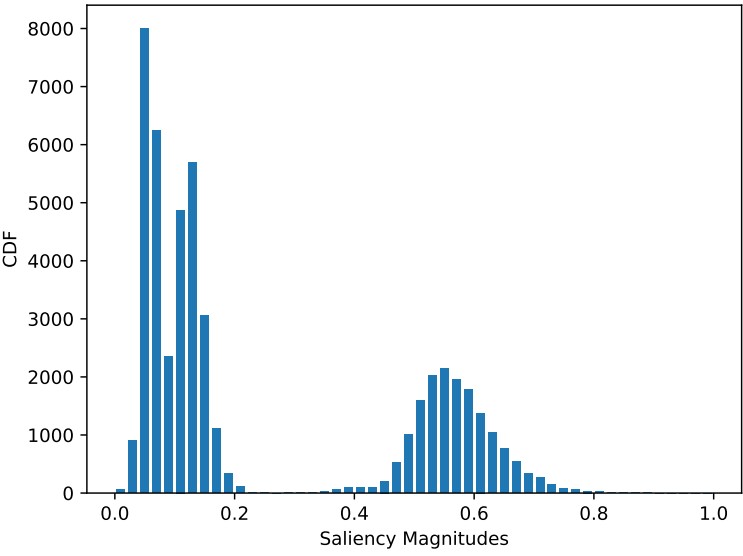

Figure 7: Distribution of adversarial saliency magnitudes for frequency components.

It is clear that there exists a large number of low-saliency (or high-robustness) frequency components with magnitudes $< 0.2$. If a majority of these components were allocated to a single model, only components with saliency magnitudes $> 0.4$ would remain for the other models. Our saliency partitioning scheme thus aims to ensure that each model in the ensemble receives a fair proportion of high-robustness components with magnitudes $< 0.2$.

The above analysis also leads us to ask — which frequency components are of high-saliency (low-robustness), and why? To this end, we visualize saliency magnitudes across all channels in Figure 8.

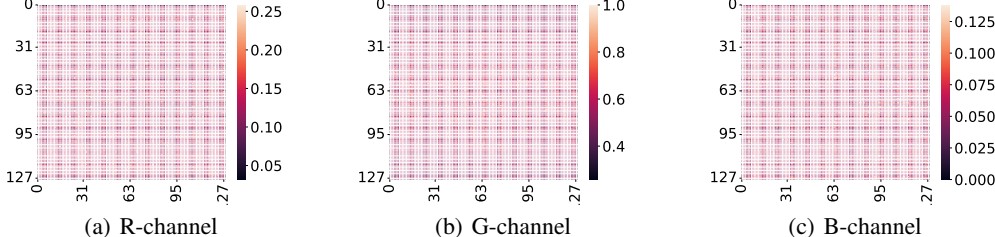

    (a) R-channel              (b) G-channel              (c) B-channel

Figure 8: Per-channel heatmaps of adversarial saliency magnitudes. Note the grid-like pattern, matching the pattern of artifacts used for deepfake detection itself.

We observe that low-saliency (high-robustness) components are largely those are constitute the grid-like artifact pattern crucial to deepfake detection itself (see Section 3.1). All remaining components appear to be high-saliency (low-robustness). Additionally, the G-channel appears to comprise the most low-robustness components.

