# OpenReview forum: "Towards Adversarially Robust Deepfake Detection: An Ensemble Approach"
_ICLR.cc/2023/Conference — Submitted to ICLR 2023_

### Official Review · Reviewer_9j5U · 2022-10-15

**Confidence:** 4
**Correctness:** 2
**Technical Novelty And Significance:** 2
**Empirical Novelty And Significance:** 2
**Recommendation:** 5

**Clarity, Quality, Novelty And Reproducibility:**

**Clarity**

The authors basically clearly present their problem and methods. The proposed method is easy to follow. But there are many ambiguous places in the experiment part, which makes the audience hard to understand.

**Quality**

The quality is overall below the line of acceptance by ICLR, with insufficient literature survey, and experiment details.

**Novelty**

The task is less explored (other than the wording "unexplored" by the authors). The proposed method is enlightened by the ensemble adversarial training but on frequency space. I suppose the technical novelty is marginal.

**Reproducibility**

Considering the inadequate details in the experiments, I suppose reproducibility is nearly impossible, without the release of code.

**Strength And Weaknesses:**

**Strength**

[1] This paper studies a less explored task, to robustify the deepfake detector. It should have practical needs and is quite important.

[2] The method is motivated by the experimental observation that redundancy exists in the frequency space, especially for deepfake images.

[3] Based on the observation, the ensemble model is natural to be considered.

[4] The authors shared some theoretical analysis of the proposed ensemble method, though I didn't check the proof exactly and am not sure if it is solid.

**Weakness**

[1] Incomplete literature survey. Missing important related work. The authors claimed "Defending against adversarial examples, in general, has been shown to be a difficult task (Athalye et al., 2018), and is unexplored in the deepfake detection setting." However, there are published papers aiming at the adversarial defense of deepfake detection. For example:

*Gandhi, Apurva, and Shomik Jain. "Adversarial perturbations fool deepfake detectors." 2020 International joint conference on neural networks (IJCNN). IEEE, 2020.*

In this IJCNN paper, the authors explored two defensive mechanisms to deepfake detectors: (i) Lipschitz regularization, and (ii) Deep Image
Prior (DIP). Thus, the statement "Defending against adversarial examples is unexplored in the deepfake detection setting" is inaccurate.

[2] Overclaim. "We propose D3, the first deep fake detection framework designed to be adversarially robust." This is an overclaim, in consideration of my last comment.

[3] Missing details in experimental settings. The descriptions on the dataset are vague. The authors claimed, "We use 50,000 images for training, 10,000 for validation, and 10,000 for testing sets". Among these, how many are real images, and how many are deepfake images? I assume it is possible that the authors first collected 70000 images from the FFHQ dataset, and then generated a deepfake image for each by StyleGAN. Also, it is not clear how to generate deepfake images by StyleGAN in this paper. The goal of deepfake detection is not clear either. Is it going to output a 0/1 binary value to verify if an image is fake or not, or output a detection region to indicate the deepfake? This relates to the understanding of the evaluation metric AUC-ROC used in this paper, but with few descriptions of its details. This brings significant difficulty in understanding the experiments and justifying the performance of the proposed method.

[4] Worse performance on FaceForensics++ dataset. In Table 8, I assume the authors are reporting accuracies (unfortunately no descriptions, and this table is even unreferred in the main text). The performance of the proposed D3 is significantly worse than the adversarial training method AT(1). Again, it seems not mentioned which types of AT the authors are using, by Madry's, or else? In a word, does this experiment indicate the proposed method only works on specific datasets, like FFHQ, but not working on video dataset FaceForensics++?

[5] This paper aims to solve a deepfake detection problem, in the adversarial setting, with datasets of human faces. But unfortunately,  almost no figures are shown in this paper to show the task. Only an image of Obama is given in Figure 1 to show the method pipeline. I am still confused after reading the whole paper, that what does the deepfake image look like? And which part of the deepfake images are detected as a clue for final judgment? Only some tables and curves are shown, but some of them lack detailed descriptions as I mentioned above.

[6] In Tables 1,2,3, plenty of 100%s are shown for the proposed method. This is very different in general image classification datasets, like CIFAR-10. It is hard for state-of-the-art adversarial defense methods to reach 60%, even with very large networks like WideResNet. I am not saying the proposed results are too good to be true, but I am curious if the authors could give some insights behind these perfect results, especially on FFHQ images. In comparison, the baselines are extremely worse, even worse than a random guess I assume. Does this indicate a stronger baseline should be considered?


**Summary Of The Paper:**

This paper focuses on learning a robust deepfake detector against adversarial attacks. The authors propose a method named Disjoint Deepfake Detection (D3). The method is motivated by the observation that redundancy exists in the frequency space, which means fewer signals are enough to make correct predictions. Based on this, the authors divide the DCT spectrum of an image into disjoint partitions and adopt an ensemble method to adversarially train each individual model. The authors show clear performance improvement, especially on the dataset of FFHQ with the deepfake generation by StyleGAN.

**Summary Of The Review:**

In summary, I vote for the rejection of this manuscript, due to the insufficient literature survey, ambiguous experiment details, and unclear result analysis.

---

> ### Author Response · Authors · 2022-11-07
> **Response to Reviewer 9j5U (part 1)**
>
> We thank the reviewer for their feedback, and address their comments below.
>
> > Incomplete literature survey. Missing important related work. The authors claimed "Defending against adversarial examples, in general, has been shown to be a difficult task (Athalye et al., 2018), and is unexplored in the deepfake detection setting." However, there are published papers aiming at the adversarial defense of deepfake detection. For example: Gandhi, Apurva, and Shomik Jain. "Adversarial perturbations fool deepfake detectors." 2020 International joint conference on neural networks (IJCNN). IEEE, 2020. In this IJCNN paper, the authors explored two defensive mechanisms to deepfake detectors: (i) Lipschitz regularization, and (ii) Deep Image Prior (DIP). Thus, the statement "Defending against adversarial examples is unexplored in the deepfake detection setting" is inaccurate.
>
> We thank the reviewer for pointing out the relevant related work, and are grateful for the pointers. We have reworded our claims to reflect the literature. The updated PDF includes our discussion of this (and other) papers, as recommended by other reviewers. To highlight the key points, we note that (i) regularization as a defense is already included in our many of our baselines, and (ii) DIP is an inference-time defense that is computationally intensive, and can take unto 30 minutes per image.
>
> > Missing details in experimental settings. The descriptions on the dataset are vague. The authors claimed, "We use 50,000 images for training, 10,000 for validation, and 10,000 for testing sets". Among these, how many are real images, and how many are deepfake images? I assume it is possible that the authors first collected 70000 images from the FFHQ dataset, and then generated a deepfake image for each by StyleGAN. Also, it is not clear how to generate deepfake images by StyleGAN in this paper. The goal of deepfake detection is not clear either. Is it going to output a 0/1 binary value to verify if an image is fake or not, or output a detection region to indicate the deepfake? This relates to the understanding of the evaluation metric AUC-ROC used in this paper, but with few descriptions of its details. This brings significant difficulty in understanding the experiments and justifying the performance of the proposed method.
>
> We apologize for missing details that affect clarity of our setup - we have now updated the PDF to include all of the requested details. To highlight the key points:
>
> (i) we use 50k, 10k, 10k fake as well as 50k, 10k, 10k real to train the classifiers.
>
> (ii) For StyleGAN, images are generated by sampling a vector from a probability distribution, and passing it through the StyleGAN generator DNN. These images represent faces of people “that do not exist”.
>
> (iii) Deepfake detection, as considered in the paper, is a binary classification task that outputs 0 or 1 (we do not consider the detection region). We have also updated the PDF with detailed explanations of AUC-ROC and adversarial accuracy as our performance metrics.
>
> We hope that the above details improve understandability of our experiments in justifying the performance of D3.
>
> > Worse performance on FaceForensics++ dataset. In Table 8, I assume the authors are reporting accuracies (unfortunately no descriptions, and this table is even unreferred in the main text). The performance of the proposed D3 is significantly worse than the adversarial training method AT(1). Again, it seems not mentioned which types of AT the authors are using, by Madry's, or else? In a word, does this experiment indicate the proposed method only works on specific datasets, like FFHQ, but not working on video dataset FaceForensics++?
>
> We thank the reviewer for this feedback - D3 does perform worse than adversarial training for deepfakes that are not generated by GANs such as FaceForensics++. The improved robustness provided by D3 relies on redundant frequency artifacts which are missing in non-GAN deepfakes. We have now updated the paper to reflect that we focus on GAN-generated deepfake images only. Appendix A.3.2 provides details for AT, i.e., Madry’s training.
>
> > This paper aims to solve a deepfake detection problem, in the adversarial setting, with datasets of human faces. But unfortunately, almost no figures are shown in this paper to show the task. Only an image of Obama is given in Figure 1 to show the method pipeline. I am still confused after reading the whole paper, that what does the deepfake image look like? And which part of the deepfake images are detected as a clue for final judgment? Only some tables and curves are shown, but some of them lack detailed descriptions as I mentioned above.
>
> We apologize for not including sufficient figures for visualization of the deepfakes, and agree that their addition would improve understandability. We have now included example deepfake images, and perturbed deepfake images in Appendix A.4.

---

> > ### Comment · Reviewer_9j5U · 2022-12-01
> > **Follow up**
> >
> > I appreciate the efforts of the authors in addressing my concerns and some of them are indeed addressed.
> > However, I believe the poor performance on non-GAN-generated deepfakes still significantly limits the applicability of the proposed method. Since in reality people hardly know the source of the deepfakes, it is unrealistic to require the potential attackers to generate deepfake by your demands. Thus I may worry about the use of the proposed method in real.
> >
> > In total, I update my rate from 3 to 5. I believe this work can still be further improved to reach the bar of ICLR.

---

> > > ### Author Response · Authors · 2022-12-04
> > > **Response to Follow up**
> > >
> > > We thank Reviewer 9j5U for their additional feedback towards improving our work. We note that FaceForensics++ contains deepfake *videos*. Alternative techniques exist to detect deepfake videos [1,2], and adversarially robust detection of deepfake videos is a separate but interesting problem (we only included evaluation for informative purposes). In general, adversarially robust detection of deepfake videos and images are both unsolved problems - our work targets the latter.
> > >
> > > Since we focus on robust detection of images, we also note that GANs are an extremely popular candidate for image generation, for purposes such as fake identities [3] and social media profiles [4].  GAN’s continue to be state-of-the-art for image generation leaderboards, per FID scores [5]. As such, a significant chunk of prior work on deepfake image detection has focused on GAN-generated deepfake images only [6-15], and many of these works have been highly cited and are influential. From an adversarial robustness perspective, detection of GAN-generated deepfake images has still remained a challenge, and we hope that addressing this segment of deepfakes is a useful contribution. Our work tries to add robustness to this area of deepfake image detection by leveraging redundancy in frequency artifacts.
> > >
> > > Nevertheless, since D3’s fundamental insight is redundancy, we note that D3 or similar technique could work well on any deepfake images where redundant features exist. Therefore, to test this hypothesis, we evaluate on a dataset of deepfake images generated using a SOTA diffusion model LDM [16]. Diffusion models have made several headlines recently with their deepfake image generation capabilities [17], and differ from GAN-based approaches. Other work indicates that diffusion models may also have frequency artifacts [18], and this suggests that D3 could work well here. We evaluate both natural and robust performance of D3 and AT on this new dataset of deepfake images. Results are below:
> > >
> > > DIffusion Model Evaluation (APGD-CE (50)):
> > >
> > > |  | Natural | &#124; L2 |  |  |  | &#124; Linf |  |  |  |
> > > | --- |--- | ---| --- | --- | --- | --- | --- | --- | --- |
> > > |  |  | &#124; 0.1 | 1 | 5 | 10 | &#124; 1/255 | 4/255 | 8/255 | 16/255 |
> > > | AT (Baseline) | 1.00 | &#124; **1.00** | 0.95 | 0.00 | 0.00 | &#124; 0.77 | 0.00 | 0.00 | 0.00 |
> > > | D3-S (4) | 1.00 | &#124; **1.00** | **1.00** | **0.01** | 0.00 | &#124; **1.00** | **1.00** | **0.01** | 0.00 |
> > >
> > > As the above table illustrates, redundant frequency artifacts can also be found in diffusion model deepfakes (as observed by the perfect natural performance), allowing D3 to offer robustness improvements over the baseline.
> > >
> > > In summary, while the primary focus of this paper is in showing that redundant features can be cleverly used to create a more adversarially robust GAN deepfake image detector, our findings on diffusion-based models suggest that the technique could be more broadly applicable. Thus, our paper opens up a new research direction for investigating adversarial robustness for broader class of deepfakes.

---

> > > > ### Author Response · Authors · 2022-12-04
> > > > **Response to Follow up (references)**
> > > >
> > > > [1] Güera, David, and Edward J. Delp. "Deepfake video detection using recurrent neural networks."  *IEEE international conference on advanced video and signal based surveillance (AVSS)*. 2018.
> > > >
> > > > [2] Li, Yuezun, and Siwei Lyu. "Exposing deepfake videos by detecting face warping artifacts." *IEEE Conference on Computer Vision and Pattern Recognition Workshops (CVPRW). 2018.*
> > > >
> > > > [3] Raphael Satter. Deepfake used to attack activist couple shows new disinformation frontier.
> > > > [https://www.reuters.com/article/us-cyber-deepfake-activist/deepfake-used-to](https://www.reuters.com/article/us-cyber-deepfake-activist/deepfake-used-to-)-attack-activist-couple-shows-new-disinformation-frontier-idUSKCN24G15E, 2020.
> > > >
> > > > [4] Paris Martineau. Facebook removes accounts with ai-generated profile photos. [https://www](https://www/).
> > > > [wired.com/story/facebook-removes-accounts-ai-generated-photos/](http://wired.com/story/facebook-removes-accounts-ai-generated-photos/), 2019.
> > > >
> > > > [5] [https://paperswithcode.com/task/image-generation](https://paperswithcode.com/task/image-generation)
> > > >
> > > > [6] Yu, Ning, et al. "Responsible disclosure of generative models using scalable fingerprinting." International Conference on Learning Representations (ICLR). 2022.
> > > >
> > > > [7] Yu, Ning, et al. "Artificial fingerprinting for generative models: Rooting deepfake attribution in training data." *IEEE/CVF International Conference on Computer Vision (ICCV)*. 2021.
> > > >
> > > > [8] Frank, Joel, et al. "Leveraging frequency analysis for deep fake image recognition." *International conference on machine learning (ICML)*, 2020
> > > >
> > > > [9] Jeon, Hyeonseong, et al. “T-GD: Transferable GAN-generated Images Detection Framework “. *International conference on machine learning (ICML)*, 2020
> > > >
> > > > [10] Bonettini, Nicolo, et al. "On the use of Benford's law to detect GAN-generated images." *International conference on pattern recognition (ICPR)*. IEEE, 2021.
> > > >
> > > > [11] Pu, Jiameng, et al. "Noisescope: Detecting deepfake images in a blind setting." *Annual computer security applications conference (ACSAC)*. 2020.
> > > >
> > > > [12] Marra, Francesco, et al. "Detection of gan-generated fake images over social networks." *IEEE conference on multimedia information processing and retrieval (MIPR)*. 2018.
> > > >
> > > > [13] McCloskey, Scott, and Michael Albright. "Detecting GAN-generated imagery using saturation cues." *IEEE international conference on image processing (ICIP)*. 2019.
> > > >
> > > > [14] Zhang, Xu, Svebor Karaman, and Shih-Fu Chang. "Detecting and simulating artifacts in gan fake images." *IEEE international workshop on information forensics and security (WIFS)*
> > > > . 2019.
> > > >
> > > > [15] Marra, Francesco, et al. "Incremental learning for the detection and classification of gan-generated images." *International workshop on information forensics and security (WIFS)*. 2019.
> > > >
> > > > [16] Rombach, Robin, et al. "High-resolution image synthesis with latent diffusion models." *IEEE/CVF Conference on Computer Vision and Pattern Recognition (CVPR)*. 2022.
> > > >
> > > > [17] Kyle Wiggers. Deepfakes for all: Uncensored AI art model prompts ethics questions. https://techcrunch.com/2022/08/24/deepfakes-for-all-uncensored-ai-art-model-prompts-ethics-questions/, 2022.
> > > >
> > > > [18] Ricker, Jonas, et al. "Towards the Detection of Diffusion Model Deepfakes." *arXiv preprint arXiv:2210.14571.* (2022).

---

> ### Author Response · Authors · 2022-11-07
> **Response to Reviewer 9j5U (part 2)**
>
> > In Tables 1,2,3, plenty of 100%s are shown for the proposed method. This is very different in general image classification datasets, like CIFAR-10. It is hard for state-of-the-art adversarial defense methods to reach 60%, even with very large networks like WideResNet. I am not saying the proposed results are too good to be true, but I am curious if the authors could give some insights behind these perfect results, especially on FFHQ images. In comparison, the baselines are extremely worse, even worse than a random guess I assume. Does this indicate a stronger baseline should be considered?
>
> We thank the reviewer for their careful analysis of our experimental results. We agree with the reviewer that SOTA for CIFAR10 struggles to reach 60%, but believe that this can be attributed to the difference in settings, i.e., deepfake detection being a binary classification task as opposed to the 10-class classification in CIFAR10. Other prior work in this area [1] has recommended AT as the recommended approach to defending against adversarial deepfakes. The poor performance of recommended baselines such as AT is predicted in Carlini et al.’s work, suggesting that our results adhere to their expectations.
>
> [1] Adversarial Deepfakes: Evaluating Vulnerability of Deepfake Detectors to Adversarial Examples
>
> > Considering the inadequate details in the experiments, I suppose reproducibility is nearly impossible, without the release of code.
>
> We apologize for this, and have now uploaded the code in a .zip.

---

> ### Author Response · Authors · 2022-11-11
> **Following up**
>
> We are grateful for the reviewer's suggestions, and hope our response was helpful in addressing the concerns. We have also updated the PDF and uploaded the source code, and are eager to answer any additional questions.

---

> ### Comment · Area_Chair_DBon · 2022-11-15
> **Feedback on authors' response**
>
> Hi Reviewer 9j5U,
>
> Please read through author's response and surface any further issue or questions to the table if any.
> If no, please also acknowledge it.
>
> Tks!

---

### Official Review · Reviewer_TsrJ · 2022-10-21

**Confidence:** 4
**Correctness:** 2
**Technical Novelty And Significance:** 2
**Empirical Novelty And Significance:** 2
**Recommendation:** 5

**Clarity, Quality, Novelty And Reproducibility:**

The paper is hard to follow due to extreme reference to the appendix and the unavailability of the important results in the main paper.

The novelty is limited as an ensemble is an active area of research and has been extensively explored utilizing frequency information.

The paper might be easy to reproduce; however, the authors have not mentioned their intention of releasing the source codes.

**Strength And Weaknesses:**

The proposed approach was found robust in handling few adversarial attacks which make the existing detection algorithms ineffective.

The claims made especially being the first work are somewhat wrong. In the literature, several works have explored the vulnerability and present robust deepfake detection approaches.

[1] EnsembleDet: ensembling against adversarial attack on deepfake detection
[2] Defending against GAN-based DeepFake Attacks via Transformation-aware Adversarial Faces

The proposed algorithm is found vulnerable against one of the adversarial attacks and hence demands extensive evaluation against state-of-the-art attacks including frequency-based attacks.

[3] Adversarial Deepfakes: Evaluating Vulnerability of Deepfake Detectors to Adversarial Examples
[4] MD-CSDNetwork: Multi-Domain  Cross  Stitched  Network for Deepfake Detection, IEEE International  Conference on Automatic Face and Gesture Recognition, 2021

The proposed algorithm should also be compared with ensemble-based approaches the baseline must also contain the results with ensemble algorithms: (i) train different networks on frequency information and (ii) utilize multiple types of input images and training of a detection algorithm.

The majority of the content in the paper refers to the appendix only, which makes the paper hard to read and follow. The authors need to make sure that all the important information is presented in the main paper.

The comparison with SOTA work is also shallow.

How the adversarial training has been performed? The results reported by AT seem wrong and need independent verification.

**Summary Of The Paper:**

The paper proposes an adversarially robust deepfake detection approach. The approaches divided the frequency information into multiple blocks based on random bisection or saliency information. This divided information is provided to the different models and later combined for robust detection accuracy. The proposed approach was found robust in handling a few attacks and acts as a baseline for future studies.

**Summary Of The Review:**

The proposed research needs attention towards the evaluation strategies including novel attacks belonging to the black box, frequency-based, and adaptive attacks not gradient-based. The comparison in the paper is shallow.

On top of that, the generalizability of the proposed algorithm against unseen adversaries also needs to be addressed.

---

> ### Author Response · Authors · 2022-11-07
> **Response to Reviewer TsrJ (part 1)**
>
> We thank the reviewer for their feedback, and address their comments below.
>
> > The claims made especially being the first work are somewhat wrong. In the literature, several works have explored the vulnerability and present robust deepfake detection approaches.
>
> We thank the reviewer for pointing out relevant related work that we missed, and greatly appreciate the pointers to references. We have revised the paper and claims to reflect this, and have addressed relevant related work in the updated PDF (see Section 2). Below we highlight how our work compares with the suggested literature:
>
> [1] EnsembleDet: ensembling against adversarial attack on deepfake detection
>
> This approach builds a traditional, standard ensemble of DNNs, each of which leverages the full pixel-space image. As such, it suffers from the same shared-features shortcoming discussed in “Ensemble Defenses” subsection of Section 2.
>
> [2] Defending against GAN-based DeepFake Attacks via Transformation-aware Adversarial Faces
>
> This paper tackles an interesting but different problem, i.e., adding perturbations to natural images to proactively prevent modification by GANs.
>
> [3] Adversarial Deepfakes: Evaluating Vulnerability of Deepfake Detectors to Adversarial Examples
>
> This paper is an attack-only paper, i.e., they investigate approaches to constructing adversarial deepfakes. In their discussion section, they “recommend approaches similar to Adversarial Training“. As per our experimental results in Section 4.2, D3 outperforms adversarial training.
>
> [4] MD-CSDNetwork: Multi-Domain Cross Stitched Network for Deepfake Detection, IEEE International Conference on Automatic Face and Gesture Recognition, 2021
>
> This paper focuses on deepfake detection in a non-adversarial context. We have now added additional references to papers that focus on such non-adversarial deepfake detection to our related work section.
>
> > The proposed research needs attention towards the evaluation strategies including novel attacks belonging to the black box, frequency-based, and adaptive attacks not gradient-based. The comparison in the paper is shallow.
>
> We agree with reviewer on importance of evaluating against SOTA attacks. We primarily evaluated D3 with AutoAttack, since to the best of our knowledge, it is an evaluation benchmark in the community. Since we attack the entire D3 pipeline, we hope gradient-based attacks would be the strongest, as they can differentiate through the pipeline. We also agree with the reviewer that it is important to consider black-box attacks, as they can reveal gradient obfuscation. We note D3 is robust to the black-box Square Attack in AutoAttack.  We thank the reviewer for their recommendation of frequency-based attacks, and have evaluated D3 against an attack from this category [1]. We find D3 stands robust upto a large $L_2 = 18.77$ - we will add this to the paper.
>
> [1] Chuan Guo, Jared S. Frank, Kilian Q. Weinberger. Low Frequency Adversarial Perturbation.
>
> > The proposed algorithm should also be compared with ensemble-based approaches the baseline must also contain the results with ensemble algorithms: (i) train different networks on frequency information and (ii) utilize multiple types of input images and training of a detection algorithm.
>
> We thank the reviewer for their recommendation of ensemble baselines. We note that our current evaluation contains results on
>
> (i) training different models on frequency information (ADP, GAL, DV columns in Tables 1, 2, 3), as we trained these baselines on the frequency space.
>
>  (ii) utilizing multiple types of input images (see Tables 6, 7 in Appendix for results with images generated by BigGAN, StarGAN).
>
> We apologize if we missed any additional baselines, but would be grateful for any additional recommendations.
>
> > The majority of the content in the paper refers to the appendix only, which makes the paper hard to read and follow. The authors need to make sure that all the important information is presented in the main paper. The paper is hard to follow due to extreme reference to the appendix and the unavailability of the important results in the main paper.
>
> We apologize for this, but are grateful for the feedback and have significantly revised the PDF to address this (e.g., moving tables to main body). We greatly appreciate any additional suggestions on content that can be moved.
>
> > The comparison with SOTA work is also shallow.
>
> We apologize that comparisons appear shallow - we did our best to ensure evaluation includes SOTA baselines (ADP, GAL, DVERGE), attacks, and deepfakes generated by GANs such as StyleGAN, BigGAN, and StarGAN. We would greatly appreciate any additional recommendations for comparison.
>
> > How the adversarial training has been performed? The results reported by AT seem wrong and need independent verification.
>
> Appendix A.3.2 presents details for adversarial training. We do our best to follow standard AT practices, and have now uploaded the code for reproducibility purposes.

---

> > ### Comment · Reviewer_TsrJ · 2022-11-24
> > **Updated Score**
> >
> > Thanks for addressing the concerns. I have updated my scores accordingly.
> >
> > I believe the paper can be further improved by adding the evaluation on black-box, transferable attacks not just on auto-attack and getting biased towards that single attack strategy.

---

> > > ### Comment · Reviewer_TsrJ · 2022-12-02
> > > **New Comments**
> > >
> > > I agree with reviewer 9j5U regarding the performance of the algorithm on the unknown source deepfake/morph detection issue. This also raises concerns about the generalizability of the algorithm against unseen attacks or variants of the generation algorithm. It is seen that the algorithms perform best on the seen dataset, and generation algorithms suffer drastically in uncontrolled settings; hence their applicability is limited.
> > >
> > > The same issue is here where the proposed algorithm is highly biased in evaluation towards one form of attack as well (auto), no unseen generation algorithms of adversaries, or deepfake types. Therefore, would like to give the rating of 4.

---

> > > > ### Author Response · Authors · 2022-12-04
> > > > **Response to Follow up**
> > > >
> > > > We thank Reviewer TsrJ for their additional feedback towards improving our work. We would like to note that AutoAttack contains 3 SOTA white-box attacks, and 1 black-box attack, but greatly appreciate the recommendation to try additional black-box attacks, and transfer attacks.
> > > >
> > > > Below we have added evaluation against multiple additional state-of-the-art white-box and black-box attacks. We use the official recommended settings for all attacks. Results are below:
> > > >
> > > > $L_\infty$:
> > > >
> > > > | Attack | Perturbation Norm | AT (baseline) | D3-S(4) (Ours) |
> > > > | --- | --- | --- | --- |
> > > > | Hop-Skip-Jump (black-box) [1] | 4/255 | **1** | **1** |
> > > > |  | 16/255 | 0.99 | **1** |
> > > > |  | 32/255 | 0.48 | **1** |
> > > > |  | 64/255 | 0 | **0.67** |
> > > > | SPSA (black-box) [2] | 4/255 | 0.46 | **0.97** |
> > > > |  | 16/255 | 0 | **0.93** |
> > > > |  | 32/255 | 0 | 0 |
> > > > |  | 64/255 | 0 | 0 |
> > > > | BIM (white-box) [3] | 1/255 | 0.99 | **1** |
> > > > |  | 4/255 | 0.015 | **1** |
> > > > |  | 8/255 | 0.005 | **1** |
> > > > |  | 16/255 | 0.005 | **1** |
> > > >
> > > > $L_2$:
> > > >
> > > > | Attack | Perturbation Norm | AT (baseline) | D3-S(4) (Ours) |
> > > > | --- | --- | --- | --- |
> > > > | Hop-Skip-Jump (black-box) [1] | 2 | **1** | **1** |
> > > > |  | 5 | 0.9 | **1** |
> > > > |  | 10 | 0.02 | **1** |
> > > > |  | 20 | 0 | 0 |
> > > > | SPSA (black-box) [2] | 10 | 0.01 | **0.92** |
> > > > |  | 20 | 0 | **0.68** |
> > > > |  | 40 | 0 | 0 |
> > > > |  | 80 | 0 | 0 |
> > > > | BIM (white-box) [3] | 0.1 | **1** | **1** |
> > > > |  | 1 | 0.99 | **1** |
> > > > |  | 5 | 0.005 | **1** |
> > > > |  | 10 | 0.005 | **1** |
> > > >
> > > > Additionally, as requested we present below black-box transfer attacks from an adversary that trains a local copy of the D3 ensemble with a random partition. Black-box transfer attacks from a full-spectrum model are already included in Appendix A.3.2. Attacks are generated using APGD-CE (50). Results are below:
> > > >
> > > > |  | Natural | &#124; L2 |  |  |  |  | &#124; Linf |  |  |  |  |
> > > > | --- |--- | ---| --- | --- | --- | --- | --- | --- | --- | --- | --- |
> > > > |  |  | &#124; 0.1 | 1 | 5 | 10 | 20 | &#124; 1/255 | 4/255 | 8/255 | 16/255 | 32/255 |
> > > > | D3-S (4) | 1.00 | &#124; 1.00 | 1.00 | 1.00 | 1.00 | 0.02 | &#124; 1.00 | 1.00 | 1.00 | 1.00 | 0.135 |
> > > >
> > > > We observe that D3 is robust against transfer attacks from a full-spectrum model (Appendix A.3.2), and from a D3 ensemble with a random partition (above table).
> > > >
> > > > **In summary, we continue to outperform the baseline under all evaluated attacks, including the newly evaluated black-box transfer attacks.** In summary, our total evaluated attacks now include:
> > > >
> > > > - White-Box: APGD-CE, APGD-CW, FAB, and BIM
> > > > - Black-box (query-based): Square, Hop-Skip-Jump, and SPSA
> > > > - Black-box (transfer): Transfer attack from a full spectrum model, and transfer attack from a random partition model
> > > >
> > > > We are grateful to the reviewer for pointing out the value of additional attack evaluation, and hope that the above results address any concerns. We are also happy to evaluate on any additional attacks for which code is available.
> > > >
> > > > Regarding additional evaluation on other deepfakes types, see our response to Reviewer 9j5U.
> > > >
> > > > --
> > > >
> > > > References:
> > > >
> > > > [1] Chen, Jordan, et al. “HopSkipJumpAttack: A Query-Efficient Decision-Based Attack”, IEEE Symposium on Security and Privacy (IEEES&P). 2020.
> > > >
> > > > [2] Uesato, O’Donoghue, et al. “Adversarial Risk and the Dangers of Evaluating Against Weak Attacks”, International Conference on Machine Learning (ICML). 2018.
> > > >
> > > > [3] Kurakin, Goodfellow et al. “Adversarial examples in the physical world”. ICLR (Workshop) 2017

---

> > > > > ### Comment · Reviewer_TsrJ · 2022-12-04
> > > > > **Response**
> > > > >
> > > > > Thanks for providing the additional results.
> > > > >
> > > > > It would be better if the authors can provide the analysis in place of just providing the numerical values as to why the proposed algorithm is working and failing of standard perturbation norm.  In other words, is the proposed algorithm effective in handling only small norms but fails as soon as the norm increases?
> > > > >
> > > > > Can we also add/show the visualization of the adversarial images to see the impact of the perturbation norm?
> > > > >
> > > > > Can we also perform analysis on the network feature maps, t-SNE plots of the features, or the decision space?
> > > > >
> > > > > What about generalizability? It is also seen that the performance of the algorithm is better on one kind of GAN image but not effective for another kind of GANs (such as Big GAN). The performance on FF++ is also low even compared to the baseline (AT).
> > > > >
> > > > > The parameters of the AT are not standard. PGD training requires a large number of iterations to be effective and multiple random restarts.

---

> > > > > > ### Author Response · Authors · 2022-12-05
> > > > > > **Response to Follow up (part 1)**
> > > > > >
> > > > > > We are glad that the additional results were helpful, and appreciate Reviewer TsrJ’s further feedback. Below we address the comments:
> > > > > >
> > > > > > > It would be better if the authors can provide the analysis in place of just providing the numerical values as to why the proposed algorithm is working and failing of standard perturbation norm. In other words, is the proposed algorithm effective in handling only small norms but fails as soon as the norm increases?
> > > > > >
> > > > > > The improvement over baseline at larger perturbation norms such as $L_\infty=0.15$ and $L_2=15$ is smaller, as these norms are so large that they affect image quality (see Figure 6, for example middle row last column there are many visible artifacts to the naked eye even at $L_\infty=0.15$ defeating the purpose of the deepfake). In other words, as the norm increases, every defense should eventually fail. Carlini et al. [1] also state this behavior is expected; attack success rate should strictly increase as norm increases. We include the results on high perturbation norms for completeness.
> > > > > >
> > > > > > Regarding the success of D3, we believe that the working of D3 against attacks (as per the tables) can be largely attributed to the reduced dimension of the adversarial subspace, which occurs due to the disjoint nature of the ensemble (see proof in Section 3.3, Section A.1). We also emphasize that D3 always outperforms or performs at the level of the baseline at both small and larger perturbation norms (see Table 1,2,3, Figure 3,4).
> > > > > >
> > > > > > > Can we also add/show the visualization of the adversarial images to see the impact of the perturbation norm?
> > > > > >
> > > > > > Examples of adversarial images to see the impact of perturbation norm can be found in Figure 6 in Appendix A.4. As observed, higher perturbation norms induce highly visible artifacts and thus are not likely to be practical deepfakes.
> > > > > >
> > > > > > > Can we also perform analysis on the network feature maps, t-SNE plots of the features, or the decision space?
> > > > > >
> > > > > > Figure 7 in Appendix A.5 show the distribution of the adversarial saliency magnitudes for the the frequency features. We can see a clear separation between two classes of features - robust and non-robust (similar to previous work [3]). Our saliency partitioning algorithm ensures each individual model in the ensemble receives a fair proportion of robust frequency features.
> > > > > >
> > > > > > Figure 8 in Appendix A.5 presents per-channel heat maps of the network frequency features, as computed using adversarial saliency magnitudes. In other words, the darker regions on these heat maps represent frequency features that are more useful for robust classification. Interestingly, these heat maps display a grid-like pattern, that aligns with the observation of grid-like frequency artifacts from prior work [2].
> > > > > >
> > > > > > > What about generalizability? It is also seen that the performance of the algorithm is better on one kind of GAN image but not effective for another kind of GANs (such as Big GAN). The performance on FF++ is also low even compared to the baseline (AT).
> > > > > >
> > > > > > Regarding performance on other GANs - we note that D3 always outperforms or performs at the level of the baseline. The improvements for StyleGAN are larger than the improvements for BigGAN and StarGAN - this can be attributed to differences in generator architectures (different number of deconvolutional layers, different strides, etc [2]).  Nevertheless, all three GANs introduce redundant frequency artifacts, allowing D3 to offer improved robustness over baseline.
> > > > > >
> > > > > > Regarding performance of FF++ — we note that this dataset consists of deepfake *videos,* which are outside the scope of our work. We only focus on deepfake images, and only included evaluation of FF++ for informative purposes. While we focused on GAN-generated deepfake images, our latest response to reviewer 9j5U indicates that D3 also works for diffusion model deepfake images.

---

> > > > > > > ### Author Response · Authors · 2022-12-05
> > > > > > > **Response to Follow up (part 2)**
> > > > > > >
> > > > > > > > The parameters of the AT are not standard. PGD training requires a large number of iterations to be effective and multiple random restarts.
> > > > > > >
> > > > > > > We thank the reviewer for their recommendations. We followed hyper-parameters for AT using the original Madry et al. paper on adversarial training [4].
> > > > > > >
> > > > > > > Regarding random restarts during training, we follow the recommendation of the original paper, which says that *“Since we are training the model for multiple epochs, there is no benefit from restarting PGD multiple times per batch—a new start will be chosen the next time each example is encountered.”*
> > > > > > >
> > > > > > > Regarding the number of iterations during training: the original work used 7 iterations for CIFAR10 (color), and 40 iterations for MNIST (black-and-white). We had selected 10 iterations to more closely align with the CIFAR10 hyper parameters. Recent prior work on adversarial training uses ≤ 10 iterations as well [5,6,7]. Nevertheless, we agree that training with more iterations would be helpful, and have evaluated additional baselines of AT with 10, 20, and 50 iterations below. AT with 20, and 50 iterations are indeed more robust than with 10 iterations. However, D3-S(4) continues to outperform all AT baselines (10,20,50), even though the individual models of D3-S(4) are only adversarially trained with 10 iterations. For example, at $L_2=10$, AT with 50 iterations achieves 0.02 accuracy, whereas D3-S(4) achieves 1.00. Similarly, at $L_\infty=16/255$, AT with 50 iterations achieves 0.00 accuracy, whereas D3-S(4) achieves 1.00.
> > > > > > >
> > > > > > > Additionally (as a side note), AT with 50 steps takes significantly more training time than D3-S(4) with 10 steps, and yet AT significantly underperforms in robustness compared to D3-S(4).
> > > > > > >
> > > > > > > |  | Natural | L2 |  |  |  | Linf |  |  |  |
> > > > > > > | --- | --- | --- | --- | --- | --- | --- | --- | --- | --- |
> > > > > > > |  |  | 0.1 | 1 | 5 | 10 | 1/255 | 4/255 | 8/255 | 16/255 |
> > > > > > > | AT - PGD 10 steps | 1.00 | 0.44 | 0.16 | 0.00 | 0.00 | 0.08 | 0.02 | 0.00 | 0.00 |
> > > > > > > | AT - PGD 20 steps | 1.00 | 0.995 | 0.99 | 0.215 | 0.01 | 0.91 | 0.81 | 0.06 | 0.00 |
> > > > > > > | AT - PGD 50 steps | 1.00 | 0.99 | 0.99 | 0.75 | 0.02 | 0.95 | 0.95 | 0.79 | 0.00 |
> > > > > > > | D3-S (4) - PGD 10 steps | 1.00 | **1.00** | **1.00** | **1.00** | **1.00** | **1.00** | **1.00** | **1.00** | **1.00** |
> > > > > > >
> > > > > > > We hope that our response helps address the reviewer’s major concerns and reviewers see the value in the paper. We thank them for the good engagement and feedback.
> > > > > > >
> > > > > > > [1] Carlini et al., “On evaluating adversarial robustness”, 2019
> > > > > > >
> > > > > > > [2] N. Yu, L. Davis and M. Fritz, "Attributing Fake Images to GANs: Learning and Analyzing GAN Fingerprints," *2019 IEEE/CVF International Conference on Computer Vision (ICCV)*
> > > > > > > , 2019, pp. 7555-7565, doi: 10.1109/ICCV.2019.00765.
> > > > > > >
> > > > > > > [3] Ilyas, Andrew, et al. "Adversarial examples are not bugs, they are features." *Advances in neural information processing systems* 32 (2019).
> > > > > > >
> > > > > > > [4] Madry, Aleksander, et al. "Towards deep learning models resistant to adversarial attacks." *arXiv preprint arXiv:1706.06083* (2017).
> > > > > > >
> > > > > > > [5] Poursaeed, Omid, et al. "Robustness and generalization via generative adversarial training." *Proceedings of the IEEE/CVF International Conference on Computer Vision*. 2021.
> > > > > > >
> > > > > > > [6] Kireev et al. “On the effectiveness of adversarial training against common corruptions”, 2022
> > > > > > >
> > > > > > > [7] Shafahi, Ali, et al. "Adversarial training for free!." *Advances in Neural Information Processing Systems* 32 (2019).

---

> ### Author Response · Authors · 2022-11-07
> **Response to Reviewer TsrJ (part 2)**
>
> > The novelty is limited as an ensemble is an active area of research and has been extensively explored utilizing frequency information.
>
> As stated in (part 1), we have added a citation to the previous ensemble-based defense against adversarial deepfakes (EnsembleDet).  We thank the reviewer for pointing us to that. We discuss this work further in (part 1) below and also discuss this in the paper.   We highlight that our work still makes the following contributions:
> (i) We observe the redundancy phenomenon in in the frequency feature-space of deepfakes, allowing the construction of performant disjoint classifiers.
> (ii) We extend Tramer et al’s work to provide bounds for adversarial subspace dimension of disjoint classifiers.
> (iii) We propose D3, a detection framework that provides SOTA adversarial accuracy against adversarial deepfakes.
>
> We could not find pointers to code for EnsembleDet to do a comparison. They only evaluated their scheme on non-GAN-generated deepfakes, which is a different application domain. Also, they did not evaluate on AutoAttack, which is usually considered the state-of-the-art adversarial attack strategy, incorporating many kinds of adversarial attacks. So, a direct comparison is not straightforward, especially without their source code. The approaches also differ philosophically in that they do not do disjoint partitioning and instead create an ensemble of multiple deepfake classifiers of different architectures that are trained on the same dataset. In contrast, we create an ensemble of classifiers with the same architecture, but that are on a disjoint set of features. That is a very different strategy. Our argument is that the concept of ensemble-based defense by itself is not new. But, how we created an ensemble is novel (based on disjoint features) and it offers advantages. Furthermore, we explore two variants of disjoint partitioning (random and saliency partitioning). This is novel work.
>
> > The paper might be easy to reproduce; however, the authors have not mentioned their intention of releasing the source codes.
>
> We apologize for not discussing the code release - we have now uploaded the code in a .zip.
>
> > On top of that, the generalizability of the proposed algorithm against unseen adversaries also needs to be addressed.
>
> We agree with the reviewer on the importance of evaluating against the strongest possible adversaries. We note that our theoretical results show a reduction in adversarial subspace dimensionality, and that such a reduction should translate to reduced adversarial volume for any attack (including unseen attacks). We hope that our AutoAttack evaluation (particularly, the white-box attacks) address this concern, but we would be grateful for any additional recommendations on attacks that we could evaluate against.

---

> ### Author Response · Authors · 2022-11-11
> **Following up**
>
> We are grateful for the reviewer's suggestions, and hope our response was helpful in addressing the concerns. We have also updated the PDF and uploaded the source code, and are eager to answer any additional questions.

---

> ### Comment · Area_Chair_DBon · 2022-11-15
> **Please give feedback**
>
> Hi Reviewer TsrJ,
>
> Please read through author's response and surface any further issue or questions.
> If no, please also acknowledge it.
>
> Tks!

---

### Official Review · Reviewer_h9Bn · 2022-10-25

**Confidence:** 3
**Correctness:** 2
**Technical Novelty And Significance:** 3
**Empirical Novelty And Significance:** 3
**Recommendation:** 8

**Clarity, Quality, Novelty And Reproducibility:**

- Clarity and quality of the presentation are good, subject to filling in the missing discussion.
- Novelty: The theoretical analysis closely follows the work of Tramèr et al. (2017).
  - I understand that the authors were mainly concerned with the problem of deepfake detection, but the frequency decomposition ensemble is fairly interesting in itself and deservers a more dedicated study, e.g., in the context of mainstream adversarial attacks.
 - I encourage the authors to continue this investigation.

- Nitpicking
  - In the notation part in Section 2, please fix $\in$ to $\subseteq$.
  - It may help to rewrite equations 2-6 in terms of the common inner expressions. Perhaps those can be called something like $\kappa_2(g_i)$ and $\kappa_\infty(g_i)$.

**Strength And Weaknesses:**

- Strengths
  - The problem is of interest and current understanding is still lacking. The proposed approach and theoretical analysis are very valuable.
  - The experiments do a good job establishing the advantage against using the full spectrum, and justify the saliency-based partitioning.

- Weaknesses
  - As pointed out by other reviewers, important discussion is missing regarding both related work and experiment setup.
  - It is not clear what is the limit of this redundancy as far as classification accuracy is concerned. At which point does the model reduce to random guessing, where real and fake appear too similar? I would like to see an experiment exploring this.

**Summary Of The Paper:**

The authors propose an ensemble approach to deepfake detection based on a frequency-domain decomposition of input images, such that each model in the ensemble is likely to see fewer non-robust features. The authors offer a theoretical justification in terms of a reduction in the dimensionality of the adversarial subspace, building upon the work of Tramèr et al. (2017). Evaluation using GAN-generated images in both the white- and black-box settings, leveraging the AutoAttack benchmark, shows significant improvements compared to SOTA.

**Summary Of The Review:**

The paper is missing important discussion of related works, acknowledging prior solutions to the same problem, as well as critical details in the experimental setup and evaluation.

Update: reviewer discussions indicate those concern are still not resolved

---

> ### Author Response · Authors · 2022-11-07
> **Response to Reviewer h9Bn**
>
> We thank the reviewer for their constructive feedback.  We address their comments below.
>
> > As pointed out by other reviewers, important discussion is missing regarding both related work and experiment setup.
>
> Thank you for pointing us to missing related work. We have addressed the missing related work in the updated PDF (see updated Section 2). We have also addressed additional details regarding our experimental setup (see updated Section 4.1). We have also uploaded the code in a .zip file for reproducibility.
>
> > It is not clear what is the limit of this redundancy as far as classification accuracy is concerned. At which point does the model reduce to random guessing, where real and fake appear too similar? I would like to see an experiment exploring this.
>
> We thank the reviewer for the suggestion of identifying the limit of redundancy. To investigate this, below we extend the analysis provided in Figure 2 of the paper to smaller subsets of the spectrum. It appears natural performance begins to drop considerably when 0.097% of the spectrum is used. Our analysis in Section 4.4 and Figure 4 indicates that such small subsets would hurt robust accuracy as well.
>
> | Subset %          | 100    | 50     | 25     | 12.5   | 6.25   | 3.125  | 1.5625         | 0.78125      | 0.390625     | 0.1953125     | 0.097656625   |
> |-------------------|--------|--------|--------|--------|--------|--------|----------------|--------------|--------------|---------------|---------------|
> | AUC-ROC (avg)     | 0.9994 | 0.9998 | 0.9998 | 0.9996 | 0.9991 | 0.9970 | 0.9884533333   | 0.9525675    | 0.890365     | 0.80545875    | 0.7164675     |
> | AUC-ROC (std dev) | 0      | 0      | 0.0001 | 0.0001 | 0.0002 | 0.0007 | 0.002983040954 | 0.0127136729 | 0.0448340314 | 0.04616281046 | 0.07171937729 |
>
> > I understand that the authors were mainly concerned with the problem of deepfake detection, but the frequency decomposition ensemble is fairly interesting in itself and deservers a more dedicated study, e.g., in the context of mainstream adversarial attacks.
>
> We appreciate that the reviewer finds the frequency partitioning approach interesting - it is a very interesting question as to whether the D3 approach would provide robustness in other domains (i.e., whether a redundant feature space exists/can be found). As a highlight of our findings - unfortunately, CIFAR10 classifiers leveraging subsets of the input feature space (pixel/frequency) do not exhibit good natural accuracies, due to a lack of redundancy in these two feature spaces. To further investigate D3's viability in CIFAR10 classification, we have now added results for experiments in which we attack a CIFAR10 D3-S4 ensemble and a CIFAR10 AT baseline to Table 9 in Appendix A.3.5. We find that while D3-S4 continues to outperform AT, the adversarial accuracies are not nearly as impressive as in the deepfake detection setting. We agree with the reviewer that this remains an interesting direction to explore.

---

> > ### Comment · Reviewer_h9Bn · 2022-11-09
> > **Appreciate the follow up**
> >
> > I will make sure to take the updates into consideration during the discussion phase.

---

> > ### Comment · Reviewer_h9Bn · 2022-11-09
> > **Revise title?**
> >
> > With EnsembleDet in the picture, would it be more appropriate to use a different title? Perhaps partitioning should be mentioned. Also, one might say that "towards" does not serve a clear purpose anymore, seeing that there is some recent work on the problem. You may also want to highlight the theoretical nature of the results. Hope you find those suggestions helpful.

---

> > > ### Author Response · Authors · 2022-11-10
> > > **Revised title**
> > >
> > > We thank the reviewer for their response, and agree with the recommendation to change the title. We also greatly appreciate the suggestions and have changed the title in the updated PDF to reflect them. We also appreciate the followup and adjusted score.

---

### Official Review · Reviewer_k4jq · 2022-10-25

**Confidence:** 3
**Clarity, Quality, Novelty And Reproducibility:** The paper seems clear and novel to me.
**Correctness:** 3
**Technical Novelty And Significance:** 3
**Empirical Novelty And Significance:** 3
**Recommendation:** 8

**Strength And Weaknesses:**

Strength

1. The idea of training ensemble of models using disjoint frequency spectrum rather than pixels conceptually makes sense. As transforming images into frequency domain can be seen as a decomposition of input features, and dispatching these features to different models can further reduce the complexity of single models (which could make them more robust).
2. The authors also provide a theoretical proof on how their method could reduce the dimension of the adversarial subspace.

Weakness

I didn’t see any major weaknesses in this paper.

**Summary Of The Paper:**

This paper proposes a method for deepfake detection against adversarial attacks. It proposes to use an ensemble of models whose inputs are subsets of frequency spectrum. It also theoretically proves that the ensemble of models reduce the dimension of the adversarial subspace, which could increase the robustness.

**Summary Of The Review:**

Overall, the paper is equipped with interesting ideas and strong experiments.

---

> ### Author Response · Authors · 2022-11-07
> **Response to Reviewer k4jq**
>
> We thank the reviewer for their positive feedback, and appreciate their vote to accept the work. We appreciate that the reviewer found that the paper has interesting ideas and thorough experiments.

---

### Author Response · Authors · 2022-11-09
**Common Response to Reviewers**

We thank the reviewers for their detailed reviews, and are very grateful for their feedback towards improving our work. After reading the reviews, we note that the key concerns were primarily: (i) missing related work, and (ii) missing experimental details including a lack of source code, (iii) novelty

To address these concerns, we made our best effort and:

(i) Added all missing references with accompanying discussion for the suggested literature in the updated PDF (now available on OpenReview). Given the related work suggested by the reviewers, we think our contributions still stand and provide more details in the updated PDF as well as in the individual responses below.

(ii) Added all requested missing experimental details, and uploaded a .zip with a README and source code for D3 for reproducibility.

(iii) Novelty: we make the following contribution: we leverage a novel redundancy observation in GAN-generated deepfakes that allows for ensembles with significantly higher adversarial accuracy, in comparison to state-of-the-art. Even when comparing with EnsembleDet (a work insightfully pointed out by the reviewers) , there are significant differences, including a novel disjoint partitioning scheme to exploit redundancy (see individual response to Reviewer TsrJ). The approaches also differ philosophically in that EnsembleDet does not do disjoint partitioning and instead creates an ensemble of multiple deepfake classifiers of different architectures that are trained on the same dataset.  In contrast, we create an ensemble of classifiers with the same architecture, but that are on a disjoint set of features. Our argument is that the concept of ensemble-based defense by itself is not new. But, how we created an ensemble is novel (based on disjoint features), and it offers theoretical and empirical advantages.

We have also provided individual responses to each of the reviewers’ comments. We hope the revisions address the concerns and lead to an improved rating for the paper.

---

### Author Response · Authors · 2022-12-10
**Common Response to Reviewers (part 2)**

We thank the reviewers for the good engagement during the second stage of discussion, and would like to summarize the outcomes from individual responses:

(i) Completed evaluation on additional white-box and black-box attacks, including black-box transfer attacks.

(ii) Completed evaluation on deepfake images from a diffusion model, to demonstrate applicability beyond GAN-generated deepfake images.

(iii) Completed evaluation against additional AT baselines trained with more iterations.

We note that D3 continues to outperform all baselines after the additional evaluation. Overall, we hope that our discussion and additional results addressed concerns and have helped improve the work.

---

### Decision · Program_Chairs · 2023-01-20

**Decision:**

Reject

**Justification For Why Not Higher Score:**

Some missing experimental evidences to support the claims.

**Justification For Why Not Lower Score:**

 Training an ensemble of models using a disjoint frequency spectrum is novel and makes sense.

**Metareview: Summary, Strengths And Weaknesses:**

The authors propose an ensemble approach to deepfake detection based on a frequency-domain decomposition of input images, such that each model in the ensemble is likely to see fewer non-robust features. The authors also offer a theoretical justification to show that the ensemble of models could reduce the dimension of the adversarial subspace and increase the robustness.

Strengths:  Training an ensemble of models using a disjoint frequency spectrum is novel and makes sense. Results show that the proposed method achieves superior results than SOTA methods. Theoretical analysis extending prior work is valuable.

Weakness: The idea of ensemble models, frequency domain, and redundancy space for adversarial robustness is not novel. Lack some experimental evidence to support the claims. The disjoint frequency spectrum plays major importance in D3, and if an adversarial attack only focuses on the original input without considering the frequency partitions, it may be unfair for pixel-based methods. Missing more details and visualisations about frequency components, e.g., How to define the d frequency components, is 64 for dct-2d? The generalization importance can also be seen from the values reported in table 6 on page 17. The evaluation is on BigGAN (type of GAN only), still, the performance is quite low as compared to the performance on other forms of GANs used in the main paper. Similar to StarGAN. That's why the evaluation under unseen GAN evaluation makes sense to showcase the effectiveness of this work. Another concern is a possible bias toward one form of adversarial attack.

**Summary Of Ac-Reviewer Meeting:**

This paper received polarised reviews. 2 reviewers incline to reject it, and 2 reviewers incline to accept it. The authors revised the paper a lot during the rebuttal period by including many additional explanations and experiment results. All agree on the merits of this paper like exploring the possibilities of an ensemble of frequency domains.
In discussion with Reviewer TsrJ, who has extensive hands-on experience in deepfake, he points out some critical flaws as below,

1. he agrees and also does not worry about the fact that the paper deals with GANs based attacks. However, as we have seen, even in the GAN space there are a lot of variations possible. This raises the concern of generalization, which is highly missing in this paper. The issue is effectively dealt with in the existing studies, which are also missing in this paper. For example, (1) Table 1 of [1] showcases different types of GANs, which can be used for different generalization evaluation settings. Other papers dealing with the freq. space have demonstrated the generalization conditions in their evaluation [2,3].

[1] CNN-generated images are surprisingly easy to spot... for now
[2] Generalizing Face Forgery Detection with High-frequency Features
[3] MD-CSDNetwork: Multi-Domain Cross-Stitched Network for Deepfake Detection

The generalization importance can also be seen from the values reported in table 6 on page 17. The evaluation is on BigGAN (type of GAN only). Still, the performance is quite low compared to the performance of other forms of GANs used in the main paper. Similar to StarGAN. That's why the evaluation under unseen GAN evaluation makes sense to showcase the effectiveness of this work.

2. Another concern is a possible bias toward one form of adversarial attack. The evaluation lacks experiments against a variety of attacks. The rebuttal highly focused on one attack, i.e., the auto attack. Similar to the GANs type variation, generalization in adversarial attack manipulation handling is a serious issue. Apart from that, we have noticed from several tables that as the strength of the perturbation increases, in the majority of the attacks, the performance degrades drastically. (Table 1-2 FAB) Table 3 also showcases another important scenario where if increase the attack steps, accuracy decreases. Applying the attacks with higher step and epsilon is crucial and standard in practice to check not only the obfuscated gradient setting but to check the resiliency. The AT setting described in A.3.1 is very weak as well and not as per the adversarial attack standard.

Table 8 also shows even the performance is significantly poor than the simple baseline of adversarial training.

3. Apart from these, why or how the proposed algorithm has achieved drastic success as compared to the baseline is still not convincing. The analysis and visualization of the feature space can be helpful in that direction.

4. The analysis of the results is also not elaborative, and hence difficult to understand why the baseline performs poorly and has seen even a 60-70% jump in the performance by the proposed algorithm.

In the rebuttal, the authors tried to explain. Still, it seems they do not resolve the primary concerns: generalizability (author also agrees that even in the seen condition, the performance is not consistent across different GANs) and even adversaries, reason of success, parameters of the baseline is still not standard.

In this case, I am inclined to reject it.